



# Using a portable FTIR spectrometer to evaluate the consistency of TCCON measurements on a global scale: The COCCON Travel Standard

Benedikt Herkommer[1], Carlos Alberti[1], Paolo Castracane[6], Jia Chen[4], Angelika Dehn[6], Florian Dietrich[4], Nicholas M. Deutscher[5], Matthias Max Frey[2], Jochen Groß[1], Lawson Gillespie[3], Frank Hase[1], Isamu Morino[2], Nasrin Mostafavi Pak[3,7], Brittany Walker[5], and Debra Wunch[3]

[1]Institute of Meteorology and Climate Research (IMK-ASF), Karlsruhe Institute of Technology (KIT), Karlsruhe, Germany
[2]National Institute for Environmental Studies (NIES), Tsukuba, Japan
[3]Department of Physics, University of Toronto, Toronto, Canada
[4]Environmental Sensing and Modeling, Department of Electrical and Computer Engineering, Technische Universität München, Munich, Germany
[5]Centre for Atmospheric Chemistry, School of Earth, Atmospheric and Life Sciences, University of Wollongong, Wollongong, NSW, 2522, Australia
[6]European Space Agency/Centre for Earth Observation (ESA/ESRIN), Frascati, Italy
[7]Now at: Institute of Meteorology and Climate Research Atmospheric Environmental Research (IMK-IFU), Karlsruhe Instute of Technology (KIT), Garmisch-Partenkirchen, Germany

**Correspondence:** Benedikt Herkommer (benedikt.herkommer@kit.edu)

**Abstract.** To fight climate change it is crucial to have a precise knowledge of Greenhouse Gas (GHG) concentrations in the atmosphere and to monitor sources and sinks of GHGs. On global scales, satellites are an appropriate monitoring tool. For the validation of the satellite measurements, and to tie them to the World Meteorological Organization (WMO) trace gas scale, ground based Fourier Transform Infrared (FTIR) networks are used, which provide reference data. To ensure the highest quality validation data, the network must be scaled to the WMO trace gas scale and have a very small site-to-site bias. Currently, the Total Carbon Column Observing Network (TCCON) is the de-facto standard FTIR network for providing reference data. To ensure a small site-to-site bias is a major challenge for the TCCON. In this work we describe the development and application of a new method to evaluate the site-to-site bias by using a remotely controlled portable FTIR spectrometer as a Travel Standard (TS) for evaluating the consistency of columnar GHG measurements performed at different TCCON stations and we describe campaign results for the TCCON sites in Tsukuba (Japan), East Trout Lake (Canada) and Wollongong (Australia). The TS is based on a characterized portable EM27/SUN FTIR spectrometer equipped with an accurate pressure sensor which is operated in an automated enclosure. The EM27/SUN is the standard instrument of the Collaborative Carbon Column Observing Network (COCCON). The COCCON is designed such that all spectrometers are referenced to a common reference unit located in Karlsruhe, Germany. To evaluate the long-term stability of the TS instrument, it is placed side-by-side with the TCCON instrument in Karlsruhe and the COCCON reference unit (the EM27/SUN spectrometer SN37, which is operated permanently next to the TCCON-KA site) between deployments to collect comparing measurements.





At each of the visited TCCON sites, the TCCON spectrometers collected low-resolution (LR) ($0.5 \, \mathrm{cm}^{-1}$) and high-resolution (HR) ($0.02 \, \mathrm{cm}^{-1}$) measurements in an alternating manner. In East Trout Lake (ETL), the TCCON spectrometer broke down while the TS was en route to the station. Hence, no side-by-side comparison was possible there. For Tsukuba and Wollongong the agreement found for $XCO_2$ is on the $0.1\%$ level. For $XCH_4$ the agreement is at the $0.2\%$ level, with the low-resolution measurements showing a low bias at both sites and for both gases. For XCO the deviations are up to $7\%$. The reason for this is likely to be an known issue with the CO a priori profiles used by TCCON over source regions.

The pressure analysis reveals excellent agreement ($0.027 \, \mathrm{hPa}$, $0.135 \, \mathrm{hPa}$, and $0.094 \, \mathrm{hPa}$) for the Tsukuba, ETL, and Wollongong sites.

## 1 Introduction

According to the 6th report of the International Panel of Climate Change (IPCC) there is overwhelming evidence concerning the human influence to the warming of the earth atmosphere (Allan et al., 2021) caused by the release of greenhouse gases (GHGs) into the atmosphere. Specifically, increasing atmospheric concentrations of $CO_2$ and $CH_4$ are the main driver of global warming. Hence, it is of utmost importance to have a precise knowledge of the GHG concentrations in the atmosphere to better quantify anthropogenic and natural sources and sinks and thus the carbon cycle. Highly accurate in-situ measurements of GHGs are performed by the ICOS-network in Europe (ICOS RI et al., 2022) and by NOAA provided in the ObsPack framework (Cox et al., 2022). In-situ measurements provide a high accuracy and precision. However, they can not directly be compared to satellite data, as satellites provide column-averaged GHG concentrations and the in-situ measurements are provided at distinct, single heights and lack representativeness on the scale of the satellite observation.. This gap can be closed by Fourier Transform Infrared (FTIR) networks which also collect column-averaged data and can be tied to the high quality in-situ measurements.

The Total Carbon Column Observing Network (TCCON) (Wunch et al., 2011a) is a collaboration of 28 (status in March 2023) FTIR spectrometer sites measuring total columns of GHGs worldwide. The official evaluation software of TCCON is called GGG, its latest version is GGG2020. For current TCCON data generated with GGG2020, the estimated error budget is $0.12\%$ ($0.47 \, \mathrm{ppm}$) for $XCO_2$, $0.22\%$ ($3.90 \, \mathrm{ppb}$) for $XCH_4$, $1.7\%$ ($1.70 \, \mathrm{ppb}$) for XCO (Column "Error budget" in Table 3 in Laughner et al. (2023b)). The absolute concentrations used to convert between an absolute and a relative error are $400 \, \mathrm{ppm}$ for $XCO_2$, $1800 \, \mathrm{ppb}$ for $XCH_4$ and $100 \, \mathrm{ppb}$ for XCO.

The site-to-site consistency for TCCON data generated with GGG2020 has be evaluated by Laughner et al. (2023b) (Table 3, column "Mean abs. dev.".) The biases are $0.11\%$ ($0.42 \, \mathrm{ppm}$) for $XCO_2$, $0.27\%$ ($4.9 \, \mathrm{ppb}$) for $XCH_4$ and $8.1\%$ ($8.1 \, \mathrm{ppb}$) for XCO. The numbers are calculated from the spread of the TCCON versus in-situ airplane profiles.

In the past the data measured by the TCCON were successfully used for satellite validation (Sha et al., 2021; Hong et al., 2022; Wu et al., 2018; Wunch et al., 2017; Yoshida et al., 2013; Wunch et al., 2011b; Dils et al., 2014) and for scientific studies like correlating the $CO_2$ concentrations in the northern hemisphere with the temperature (Wunch et al., 2013) or for evaluating the biosphere exchange (Messerschmidt et al., 2013).



In order to produce reliable reference data, it is important to ensure that the network as a whole is accurately tied to the World
Meteorological Organization's (WMO) trace gas scale, and that the network has minimal station-to-station biases. Currently,
this connection to the WMO trace gas scale is achieved by vertically integrating collocated profile observations by airborne
profile observations or via a new technique called AirCores (Karion et al., 2010) to compare with the TCCON results (Wunch
et al., 2010; Messerschmidt et al., 2011; Sha et al., 2020). In short, AirCore profiles are derived by mounting a long, evacuated
tube on a balloon or aircraft. During descent, the tube gets filled. Height resolved profiles of GHG concentrations can be derived
from the record.

However, the collection of such a profile data set is laborious, expensive and the number of available in-situ profiles is too
small for detecting minor biases of individual TCCON sites. Moreover, TCCON sites located in populated regions with severe
flight restrictions are particularly difficult to address with this strategy.

In addition to the in-situ comparisons, the TCCON quality assurance (QA) has two supplementary methods: The monitoring
of the instrumental line shape (ILS) and the evaluation of XAIR (or XLUFT). They are explained in detail in Section 2.1. In
short, both methods are used to detect deviations of the instrument from its optimal behavior. However, while both the ILS anal-
ysis and the XAIR evaluation are very useful methods for detecting deviations from the expected instrumental characteristics
at individual sites, they cannot guarantee that the final XGas products will be consistent within the network.

In this work an additional method of further enhancing the TCCON's quality management is presented and applied. It is
based on a portable EM27/SUN FTIR spectrometer operated in the framework of the Collaborative Carbon Column Observ-
ing Network (COCCON) (Frey et al., 2019) which will be used as a traveling standard. The EM27/SUN spectrometer is a
low-resolution, portable FTIR spectrometer. The prototype was developed by the Karlsruhe Institute of Technology (KIT) in
cooperation with Bruker starting in 2011 (Gisi et al., 2012) and became available as a commercial item in 2014. In 2015 an
extension of the original configuration was implemented by adding a second detector covering the 4000 - 5000 $\mathrm{cm}^{-1}$ spectral
range (Hase et al., 2016). This additional channel allows to retrieve XCO and an alternative $XCH_4$ product, which we refer to
as $XCH_4^{S5P}$, as the same spectral region is measured to retrieve $CH_4$ by the space borne TROPOMI spectrometer onboard the
Sentinel 5P (S5P) satellite.

The EM27/SUN spectrometer has proven its high level of instrumental stability in various city campaigns (Tu et al., 2022;
Alberti et al., 2022b; Hase et al., 2015; Dietrich et al., 2021; Chen et al., 2016) and long-term studies (Alberti et al., 2022a). It
has even been successfully deployed on ships (Klappenbach et al., 2015; Butz et al., 2022) or on cars (Butz et al., 2017; Luther
et al., 2019).

Due to the stable instrumental characteristics it is meaningful to perform side-by-side comparisons of EM27/SUN spectrom-
eters to quantify residual instrument specific imperfection in the framework of campaign deployments. Moreover, this finding
enables the COCCON to evaluate all EM27/SUN FTIR spectrometers before the first deployment and thereby connecting all
spectrometers to a common reference (Alberti et al., 2022a; Frey et al., 2019).





Local campaigns for comparing subsets of TCCON sites have been performed using EM27/SUN spectrometers (Mostafavi Pak et al., 2023; Hedelius et al., 2016). Here we present the commissioning and the first results achieved with a dedicated Travel Standard (TS) unit for systematically evaluating the station-to-station consistency of the TCCON on a intercontinental scale.

Karlsruhe is chosen as the home-base of the TS. This is the natural choice as in Karlsruhe there is a TCCON site as well as the reference EM27/SUN spectrometer for the whole COCCON network. Hence, the TS is calibrated against the COCCON reference and the Karlsruhe TCCON site.

Physically, the TS is an EM27/SUN spectrometer housed in an enclosure enabling autonomous operation (Heinle and Chen, 2018; Dietrich et al., 2021). The unit is equipped with a high accuracy pressure sensor (Vaisala PTB330, Vaisala (2023)). By

comparing the TCCON sites to a common reference this approach allows us to compare the TCCON spectrometers to each other. Here, we use side-by-side measurements of the TS with the TCCON sites to compare the XGas results. For this it is important to note that the TS is a low-resolution spectrometer and that XGas results derived from spectra recorded side-by-side with different spectral resolution can differ due to various causes. This is examined in Petri et al. (2012) and also described in Section 2.2.2.

In order to avoid the resulting uncertainties connected to differing resolution, additional low-resolution double-sided interferograms are recorded with the TCCON spectrometer and these are used in addition to the high-resolution TCCON measurements for the side-by-side comparison. Note that, due to the lower resolution of the TS, its interferograms are lacking the high-resolution section of the interferograms recorded by the TCCON instruments. Therefore, it is not possible to fully evaluate the performance of a TCCON spectrometer by comparison with the TS. However, the gas cell measurements performed by

TCCON cover this missing aspect of verifying the high-resolution part of the TCCON interferogram by providing a characterization of the ILS. A more detailed description of the procedures for measuring station-to-station consistency is provided in the following sections.

The paper is structured as follows: After this introductory section, the second section introduces the idea and the design choices as well as the practical realization of the TS. The third section describes the procedure of monitoring the TS spectrom-

eter by laboratory and side-by-side reference measurements performed at KIT between the campaigns. In the fourth to sixth section, the data resulting from the observations collected with the TS and the TCCON station spectrometer in Japan, Canada and Australia are presented. The seventh section presents quantitative comparisons between the visited sites and the COCCON reference spectrometer operated in Karlsruhe. The eighth section gives a summary and an outlook.

## 2 The Travel Standard: Idea and Realization

### 2.1 Idea and Description of the Travel Standard

The creation of a TS originates from the desire to detect potential station-to-station biases across the TCCON on a global scale. The most direct approach to solve this would be to collect side-by-side measurements of the FTIR-spectrometers in the TCCON. Unfortunately, the spectrometer used by the TCCON are large, heavy and sensitive, so shipping them around is challenging. More importantly, the instrumental characteristics of the IFS125HR spectrometer, used as the standard TCCON



spectrometer, can not be kept stable during transportation, as a partial dismounting of the interferometer is required for safe transport and variable loads occurring during transport disturb the previous alignment state.

In the past, side-by-side measurements with different TCCON spectrometers have been attempted by several investigators, and these encounters were very useful for gaining insights which helped to further improve the performance of TCCON (Pollard et al., 2021; Messerschmidt et al., 2010). While these studies demonstrated the typical level of consistency achievable
in practice with IFS125HR spectrometers, they do not provide an actual side-by-side check of two TCCON sites.

Instead, there are several network-wide consistency checks checks as outlined in the introduction which are:

– Comparison with height resolved in-situ data collected by airplanes and AirCores.

– The evaluation of the ILS.

– The evaluation of XAIR.

In the following, these methods are described in more detail than in the introduction and their limitations are discussed. Further technical quantities which are side results of the spectral fits (as, e.g., abscissa wavenumber scale or stretch of the solar absorption lines contained in the spectrum) are also used for QA/QC of the TCCON data products but are not discussed further.

**Comparison with in-situ data:** So far, the TCCON has used in-situ measurements collected by airplanes or balloon-based AirCores to assess site-to-site consistency, and tie the TCCON measurements to the WMO trace gas standard scale (Wunch
et al., 2010; Messerschmidt et al., 2011; Karion et al., 2010; Sha et al., 2020).

However, those measurements are sparse, infrequent and difficult to conduct in highly populated areas with dense air traffic. Nevertheless, they are important for tying TCCON as a whole to the WMO scale, and they can contribute to the performance assessment of individual sites.

**ILS evaluation:** The use of a gas cell for evaluation of the ILS was implemented for the Infrared working group (IRWG) of
the Network for the Detection of Atmospheric Composition Change (NDACC) in the 1990s, for details see Hase et al. (1999). The cell is filled with a known amount of a target gas at low pressure and the ILS is deduced from the comparison of a measured spectrum with a simulated spectrum using the known cell characteristics (length, pressure, temperature).

The measurements offer high sensitivity for detecting deviations of the spectrometer's modulation efficiency as function of the optical path difference (OPD) from nominal behavior. The procedure essentially ensures that the shape of spectral lines in
the measured atmospheric spectra is reproduced properly.

This procedure however, covers only a limited spectral range where the cell gas offers useful spectral signatures. Low-pressure gas cells mainly provide a check of the ILS for a high-resolution spectrometer. To verify the modulation efficiency near zero path difference, which is relevant for the quantification of tropospheric species, additional cells containing gas mixtures at higher pressure would be useful (Hase, 2012). But the preparation and use of different cells is laborious and has
not yet been implemented in the operational procedures of the TCCON or the NDACC FTIR networks. Moreover, it is less sensitive for detecting minor disturbances of the low-resolution part of the spectrum (at low OPDs) or for validating the zero level baseline of the recorded atmospheric spectra. Such disturbances critically affect the measured line area and thereby the derived column-averaged GHG concentrations.



**XAIR calculations:** XAIR is a parameter calculated by the retrieval algorithms to check for consistency. In GGG it is
implemented as (Wunch et al., 2015),

$$\mathrm{XAIR} = \frac{VC_{\mathrm{air}}}{VC_{\mathrm{O_2}}} \cdot 0.2095 - X\mathrm{H_2O} \cdot \frac{m_{\mathrm{H_2O}}}{m_{\mathrm{dry-air}}}, \tag{1}$$

$$VC_{\mathrm{air}} = \frac{p_s}{\bar{g} \cdot \frac{m_{\mathrm{dry-air}}}{N_A}} . \tag{2}$$

Here, $VC_{\mathrm{O_2}}$ is the total number of $O_2$ molecules in the air-column, $X\mathrm{H_2O}$ the column-averaged, dry air mole fraction of
$H_2O$, $m_{\mathrm{H_2O}}$ and $m_{\mathrm{dry-air}}$ are the mean molar masses of $H_2O$ and dry air, respectively, $N_A$ is Avogadro's constant and $\bar{g}$
the column-averaged gravitational constant. The first part in (1) compares the total column of dry air ($VC_{\mathrm{dry-air}} = \frac{VC_{\mathrm{O_2}}}{0.2095}$) to
the amount of air molecules calculated by using the surface pressure and assuming a hydrostatic balanced atmosphere. The
surface pressure however, depends on the amount of water vapor in the atmosphere. This is considered in the second term.
As a technical quantity it is created to deliver a value near unity for a spectrometer correctly set up and aligned. According to
Laughner et al. (2023b) for the TCCON the expected value is 0.999 due to imperfections in the $O_2$ spectroscopy.

Deviations from this expected value indicate an error with the instrument. Known causes are a bad instrumental line shape
(ILS), nonlinearity at the detector, sampling ghosts, an error in the used surface pressure measurement, in the spectroscopic
measurement, or in the estimation of airmass (e.g. line of sight not properly centered on solar disc, undetected time offset). In
this work the data are also evaluated with PROFFAST2, which is the official retrieval software of the COCCON community
(Hase et al., 2023; Feld et al., 2023). It is developed at KIT and is explicitly designed to be used with EM27/SUN spectrom-
eters, however, it is also able to handle measurements of several other FTIR low-resoulution instruments. When comparing
XAIR values of GGG and PROFFAST it is important to note that the implementation in both packages is inverse to each
other. Consequently, when in this paper XAIR of PROFFAST and GGG are compared to each other, the value calculated by
PROFFAST is inverted.

However, both, the cell measurements and the XAIR methods do not explicitly validate the final XGas products. Hence it is
not possible to guarantee the compatibility of XGas data sets collected by different stations based on the cell methods and the
XAIR quantity.

We therefore believe that the COCCON-TS for the TCCON presented in this paper is a valuable complement to the methods
presented above: The TS uses exactly the same measurement principle as the TCCON and the retrieved XGas values can be
compared directly to each other. The TS is easily transportable and is independent of potential overflight restrictions affecting
airplane or AirCore measurements. In addition, it is a reasonably inexpensive activity as the measurements can be collected
remotely, assuming support of the local TCCON staff. The costs are dominated by shipping. A practical limitation is that
temporary import of the TS into countries not recognizing the ATA carnet (a possibility to tax and duty free temporary import
and export of scientific goods) agreement is more difficult to achieve.



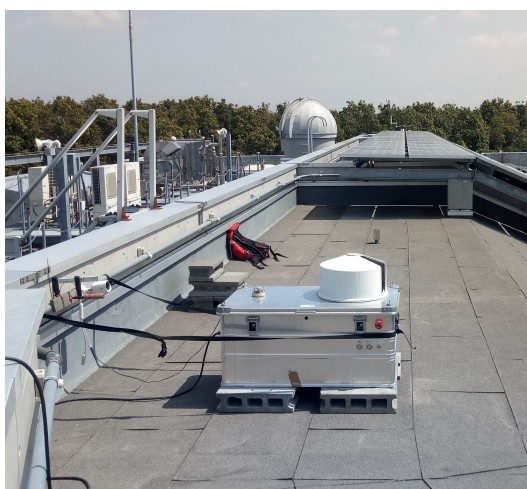

**Figure 1.** The figure shows the TS in Tsukuba, Japan whilst measuring. The enclosure including its measurement dome, developed by TU Munich, can be seen in the foreground. The white hemisphere in the background is the TCCON dome.

## 2.2 Shelter Hardware and Standardized Procedure

### 2.2.1 Shelter Hardware

For a TS based on a EM27/SUN spectrometer there are two key demands. The first is that it needs some kind of enclosure which helps to make the field deployment at the various sites simple and controllable remotely. As it protects the EM27/SUN from precipitation it is not necessary to deploy it manually for each measurement day which helps to collect more observations. The second is that the hardware should help to maintain temperature and humidity inside the shelter within a range that allows

the spectrometer to operate under a wide range of ambient conditions. This is realized by using an enclosure which was developed by TU Munich (Heinle and Chen (2018); Dietrich et al. (2021)). It is equipped with an easy to use and reliable software running on a programmable logic controller to control the measurement dome and an internal computer to control the EM27/SUN spectrometer. Figure 1 shows the enclosure including the rotatable dome. Remote access is provided by a router which can connect to the internet via LAN, Wi-Fi, or even cellular data. To provide stable temperature and humidity conditions

the enclosure is equipped with a heater and a fan to heat and cool the inside of the enclosure depending on ambient conditions. The temperature is kept above 25 °C to prevent condensation. In a hot summer day, the maximal temperature measured was 40 °C, which is in a range the EM27/SUN spectrometer operates without problems. A rain sensor is mounted to the cover which, in case of rain, induces a rapid closing of the dome to protect the EM27/SUN spectrometer. A small UPS is included to close the dome in case of a blackout to not leave the spectrometer unprotected. Since the enclosure was primarily designed to

be used in Europe, it was in its original configuration not able to deal with power grids other than the European one. Hence, the enclosure was modified at KIT to enable the use with different voltages and frequencies of power grid all over the world. To accurately retrieve XAIR and XGas values, the precise knowledge of the surface pressure is crucial. A study of Tu (2019)



using PROFFIT as an evaluation software with low-resolution spectra showed that a change of 1 hPa in the measured ground pressure causes an average increase of about 0.035% in $XCO_2$, 0.039% in $XCH_4$ and 0.052% in XCO, respectively. The

TCCON data protocol requires a pressure uncertainty of maximum 0.3 hPa. To measure this important variable, the enclosure was equipped with a Vaisala PTB 330 meteorological pressure sensor. Its accuracy is given as 0.1 hPa (Vaisala, 2023) and is therefore accurate enough for comparing the pressure of the TCCON sites.

Furthermore, two transport loggers (ASPION G-Log2) are added to monitor temperature and humidity during the shipping and to detect the occurrence of mechanical shocks. The loggers are attached to the enclosure as well as to the EM27/SUN

directly. The EM27/SUN is transported in a separate box and packed in foam. The loggers are saturated at 16 g. Hence, all shock events larger than that are truncated to 16 g.

At the shipments for the campaigns in Tsukuba and ETL no shock events were recorded for both sensors. At the shipment towards Wollongong, the logger attached to the spectrometer recorded one even with a maximum acceleration of 16 g. From Wollongong to Karlsruhe two events with 16 g peak were recorded. The logger attached to the enclosure recorded three shock

events on its way to Wollongong (with maximum accelerations of 8.8 g, 14.8 g, 16 g and 16 g) and one shock event (maximum of 16 g) on its way back to Karlsruhe.

It is important to note that the records of the logger attached to the spectrometer were recorded just a few minutes after the record have been started or before it was stopped, respectively. In retrospect, it is not possible to tell if the logger was already attached to the spectrometer when the event was recorded, or if the events were the result of careless handling of the logger

when carrying it around. Nevertheless, the records show, that the TS went through rough conditions at the shipments of the Wollongong campaign as it experienced shocks up to 16 g. As a comparison, the maximum acceleration of the Saturn V rocket was at maximum 3.8 g (Figure 4-3 in NASA (1969)).

To sum up this section, the TS comprises the EM27/SUN itself but also the enclosure including the pressure sensor and the transport logger.

**2.2.2 Procedure**

To perform measurements as consistently as possible, the same procedure is used at each site. In addition, before and after each visit, the TS device is sent back to KIT, where solar measurements are collected next to the COCCON reference device which is operated continuously near the TCCON site in Karlsruhe. Furthermore, laboratory measurements (open path and gas cell measurements) are performed. The solar and laboratory measurements are described by Frey et al. (2015) and Alberti

et al. (2022a). These tests are used to monitor the spectrometer between the campaigns to identify any potential errors like misalignment or damages at that sun-tracker that may have caused by shocks during the transportation. Furthermore, the transport logger, which monitors acceleration, temperature and relative humidity, is read out and it is checked if its recorded values are in a critical range.

At the TCCON sites several days of side-by-side measurements are performed. During the visit, care is taken that the TCCON

measurements procedure collects alternating high-resolution measurements with the operational TCCON settings (single sided



interferograms (IFGs) with a maximum optical path difference (MOPD) of mostly 45 cm) and low-resolution measurements matching the spectral resolution of the EM27/SUN spectrometer (double sided IFGs with a MOPD of 1.8 cm).

The resolution of the instrument can induce deviations in the XGas values due to the following reasons: (1) differing vertical sensitivities. These generate XGas differences if the a-priori vertical profile shape of the gas deviates from the actual profile.
(2) Residual deviations of modulation efficiency at large OPD (affecting the spectrometer used to collect the high-resolution spectrum); (3) different error propagation into the XGas result in the presence of other disturbances, e.g.; channeling (resonances due to an unintended cavity in an optical element, see e.g. (Frey, 2018)); (4) different error propagation into the XGas derived from either single-sided and double-sided interferograms in presence of residual phase errors. Double-sided interferograms allow for a superior photometric accuracy (Davis et al., 2001). Hence, the low-resolution measurements are recorded
to ensure that no resolution-induced effects influence the comparisons. Another advantage of running the TCCON instruments at lower resolution is that it allows us to process the IFGs in an identical fashion as for the EM27/SUN spectrometer's IFGs with the PROFFAST2 retrieval software. This results in a data product collected with the IFS125HR which comparable to the EM27/SUN spectrometer measurements.

Both, the TCCON and the PROFFAST retrieval algorithms scale an a-priori profile to retrieve the $\mathrm{XGas}$ values. To avoid
biases between COCCON and TCCON results due to the usage of different a-priori profiles, the COCCON retrieval performed by PROFFAST2 uses the same a-prioris as the TCCON.

For the visit at each site three aims can be identified. Foremost, the comparison of the low-resolution spectra of the TCCON site and the EM27/SUN spectrometer is used to search for any instrumental issues.

In addition, any biases between the official TCCON product and the COCCON product derived from the TS measurement
can be evaluated. Individual results collected over the course of a few days, however, need to be treated with care, as any imperfection of the a-priori trace gas profiles will induce differences in the XGAS results due to the different vertical sensitivities (compare with Section 4). Nevertheless, after having performed a larger number of site visits, it will become possible to statistically analyze the individual results and to quantify the systematic biases with increasing confidence.

Finally, the $\mathrm{XAIR}$ and pressure measurements of each TCCON site are compared with the measurements collected by the
TS.

For the comparison of the two instruments it is necessary to calculate the observed bias between the two instruments. This is realized by using so-called bias compensation factors $K_{\mathrm{B}}^{\mathrm{A}}$: Assuming $\overline{\mathrm{XGas}_{\mathrm{A}}}$ and $\overline{\mathrm{XGas}_{\mathrm{B}}}$ are the time-averaged XGas measurements of instrument $A$ and $B$ the bias compensation factor describes the instrument-to-instrument bias by $\overline{\mathrm{XGas}_{\mathrm{A}}} = K_{\mathrm{B}}^{\mathrm{A}} \cdot \overline{\mathrm{XGas}_{\mathrm{B}}}$. The procedure to calculate them is given in Appendix A. Before calculating the bias compensation factors, the
data are filtered by the following criteria:

1. The preprocessor of PROFFAST2 checks for variations and the mean of the DC level of the interferogram which indicates clouds or a poor tracking. The mean DC level is calculated by first smoothing the recorded interferogram using a rolling mean and then taking the average of the smoothed data. The DC variation is calculated by taking the quotient of the absolute maximum and the absolute minimum of the smoothed data and subtracting one. All interferograms with a mean





       DC level smaller than 0.5 and a DC variation larger than 0.1 are rejected. These numbers are the default settings as given in the templates of PROFFAST2.

2. All data recorded at a solar zenith angles (SZA) larger than 80° are filtered out and removed from the comparisons. This is because at larger SZA, the airmass varies faster. The larger the airmass the larger are the impacts of spectroscopic inaccuracies which increases the measurement uncertainties. In addition, empirical airmass-dependent corrections and the assumption of hydrostatic balance become less reliable.

3. Measurements with obvious outliers in XAIR are deleted. They are determined by calculating the standard deviation $\sigma_{\mathrm{XAIR}}$ of XAIR for each day. All data points outside of $2 \pm \sigma_{\mathrm{XAIR}}$ are assumed to be outliers and thus deleted.

4. Last, all remaining obvious outliers for each species are deleted as well. The limits upper-lower limits used for this are 1.6 - 1.95 ppm for $XCH_4$, 350 - 450 ppm for $XCO_2$ and 40 to 200 ppb for XCO.

All data shown in the figures in this paper and used for calculation are filtered as described above.

## 3   Results of the TS characterization at KIT and empirical biases monitoring between the campaigns

The COCCON XGas units are tied to the TCCON via the COCCON reference EM27/SUN spectrometer (serial number SN37) which is operated continuously at KIT next to the Karlsruhe TCCON site. The multi-annual XGas data resulting from the PROFFAST2 analysis of SN37 is bound to match with the Karlsruhe TCCON station by airmass independent correction factors (AICF) as well as by airmass dependent correction factors (ADCF). These factors are implemented in PROFFAST2 accordingly. For the retrievals with PROFFAST2, the calibration released with the PROFFASTpylot tag 1.2 (Feld et al., 2023) is used.

To monitor the TS instrument the same procedure is used: Before and after each campaign the TS instrument (serial number SN39) is compared to the reference EM27/SUN spectrometer by collecting side-by-side measurements. These measurements are used to determine the instrument bias compensation factors $K_{\mathrm{SN39}}^{\mathrm{SN37}}(\mathrm{XGas})$ for $XCO_2$, $XCH_4$ and XCO. These factors are used to check if the TS instrument misaligned during the campaigns (especially due to the shipments).

In Figure 2 the XGas values of the side-by-side measurements are plotted, with the data of the reference instrument plotted in blue and the TS data in orange. All the measurements were collected in Karlsruhe between the campaigns for two days each: Before the Japan campaign in December 2021 and January 2022, between the Japan and Canada campaigns in June 2022, between the Canada and Australia campaign in October 2022 and after the Australia campaign in March 2023.

A visual inspection reveals a good agreement and stable results for XAIR, $XCO_2$ and $XCH_4$ during all four measurement periods. For XCO, however, there is a larger difference in the second period (collected in Karlsruhe between the Japan and the Canada campaign), which is reduced again in the third period (collected in Karlsruhe between the Canada and the Australia campaign). A closer investigation for this behavior is given in Section 3.1, where an empirical correction for the variable XCO bias is derived. This correction is applied to the data of the TS spectrometer and plotted using the red crosses in the figure. The increased noise levels (2021-12-22, 2022-06-02, 2023-03-16, 2023-03-22) are likely due to cloudy weather on these days.





This results in higher DC variations of the interferograms and reduced quality of the solar tracking. Due to tight schedule it was necessary to also use non-perfect weather conditions.

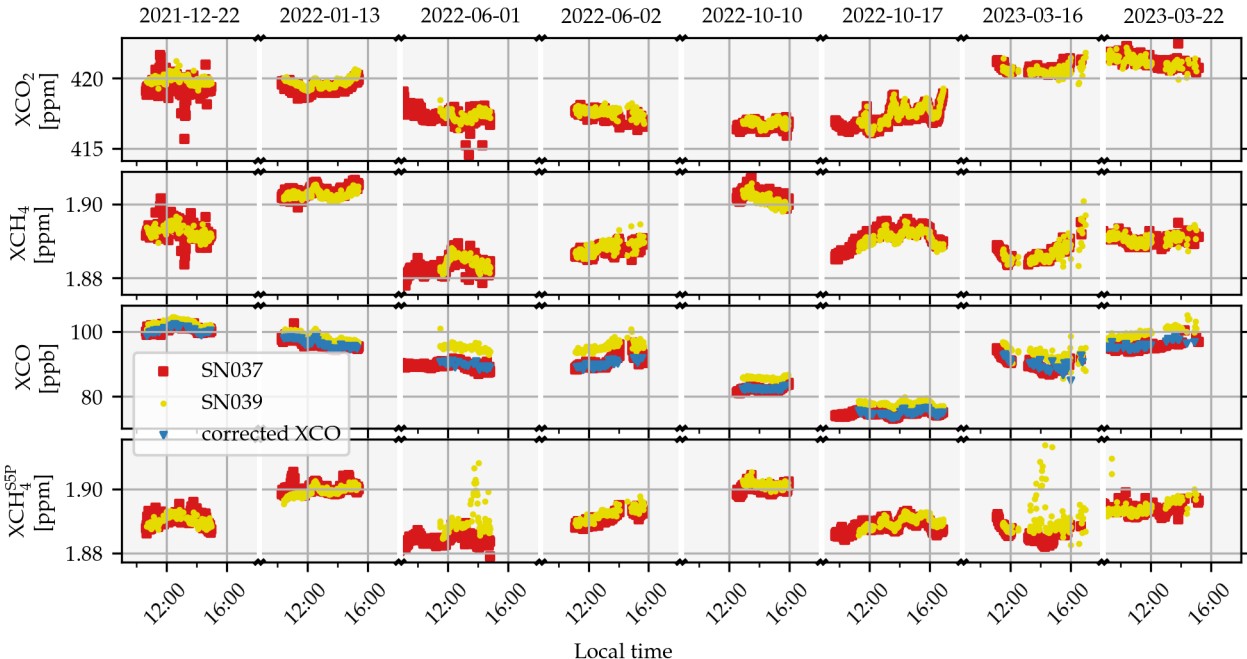

**Figure 2.** The result of the side-by-side measurements of the COCCON reference device in blue and and the TS in orange. In the top panel $XCO_2$ is plotted, in the mid panel $XCH_4$ and XCO in the lowest $XCH_4^{S5P}$. For each of the gases empirical bias compensation factors are calculated and summed up in Table 1. For $XCO_2$ and $XCH_4$ the correction show minor variability over time. For XCO, however, there is a significant variability. This variability is corrected using an ad-hoc empirical solar zenith angle dependent function. The corrected data are plotted using the red "x"-markers. From the corrected XCO data only every 12[th] marker is plotted to provide a clearer figure. For more details refer to Section 3.1.

The bias compensation factors $K_{SN39}^{SN37}$ are calculated and summarized in Table 1. For XCO, the corrected data are used to calculate the bias compensation factors. The errors are calculated using the procedure described in Section A1. Furthermore, the table shows the relative deviation of the correction factor to the row above (i.e. to the previous deployment in KA) $\Delta K_{SN39}^{SN37}$ in percentage. In addition, for each measurement period the temporal mean of all XGas values is calculated for both, the reference and the TS instrument and the difference $\Delta\overline{XGas}$ is calculated. The change in this quantity relative to the previous factor is given as $\Delta(\Delta\overline{XGas})$. The relative change of the correction factors in percentage $\Delta K_{SN39}^{SN37}$, as well as $\Delta(\Delta\overline{XGas})$ are used to check the stability of the two instruments.

The absolute change in the temporal mean values for all gases is less than the estimated site-to-site biases of the TCCON given in the introduction. From this it can be seen that the stability of the TS EM27/SUN spectrometer is good enough for comparing TCCON stations.





**Table 1.** Tabulated bias compensation factors for the comparison of the TS spectrometer unit with the reference instrument. The bias compensation factors $K_{\text{SN39}}^{\text{SN37}}$ are calculated using the data showed in Figure 2. For XCO the corrected values (red crosses) are used. For more details on this please refer to the main text. $\Delta K_{\text{SN39}}^{\text{SN37}}$ [%] denotes the deviation to the correction factor in the row above. $\Delta\overline{\text{XGas}}$ denotes the difference of the temporal mean over each measurement period. $\Delta(\Delta\overline{\text{XGas}})$ denotes the change of the difference to the previous encounter. For a evaluation of the stability of the instruments the values of $\Delta K_{\text{SN39}}^{\text{SN37}}$ and $\Delta(\Delta\overline{\text{XGas}})$ are the important values. The values in % for $\Delta(\Delta\overline{\text{XGas}})$ are given for a direct comparison with the estimated TCCON site-to-site consistency (Laughner, 2023). To convert from the mixing ratio to percentage we used 400 ppm for $XCO_2$, 1800 ppm for $XCH_4$ and 100 ppb for XCO. The smaller the $\Delta(\Delta\overline{\text{XGas}})$, the more stable the instruments are against each other. For all instrument the drift between two characterization measurements is less than the accuracy estimated for TCCON.

| Species | Date | $K_{\text{SN39}}^{\text{SN37}}$ | $\Delta K_{\text{SN39}}^{\text{SN37}}$ [%] | $\Delta\overline{\text{XGas}}$ | $\Delta(\Delta\overline{\text{XGas}})$ | estimated TCCON accuracy |
|---|---|---|---|---|---|---|
| XCO2 | January 2022 | $0.99887 \pm 0.00004$ | − | −0.4684 ppm | − | 0.2 % |
|  | June 2022 | $0.99942 \pm 0.00007$ | 0.06313% | −0.2575 ppm | 0.21096 ppm (0.053 %) |  |
|  | October 2022 | $0.99960 \pm 0.00003$ | 0.01228% | −0.1626 ppm | 0.09484 ppm (0.024 %) |  |
|  | March 2023 | $1.00036 \pm 0.00005$ | 0.07060% | 0.1444 ppm | 0.30700 ppm (0.077 %) |  |
| XCH4 | January 2022 | $1.00036 \pm 0.00004$ | − | 0.0007 ppm | − | 0.43 % |
|  | June 2022 | $0.99962 \pm 0.00006$ | −0.06684% | −0.0006 ppm | −0.00129 ppm (−0.072 %) |  |
|  | October 2022 | $1.00066 \pm 0.00002$ | 0.09862% | 0.0013 ppm | 0.00188 ppm (0.104 %) |  |
|  | March 2023 | $1.00004 \pm 0.00005$ | −0.07077% | −0.0001 ppm | −0.00135 ppm (−0.075 %) |  |
| XCO | January 2022 | $1.00159 \pm 0.00029$ | − | 0.1608 ppb | − | 5.4 % |
|  | June 2022 | $1.00071 \pm 0.00075$ | −0.05360% | 0.0831 ppb | −0.07767 ppb (−0.078 %) |  |
|  | October 2022 | $1.00052 \pm 0.00022$ | −0.04768% | 0.0403 ppb | −0.04282 ppb (−0.043 %) |  |
|  | March 2023 | $0.96076 \pm 0.00054$ | −0.58717% | −0.4636 ppb | −0.50394 ppb (−0.504 %) |  |
| $XCH_4^{\text{S5P}}$ | January 2022 | $1.00036 \pm 0.00003$ | − | 0.0006 ppm | − | N/A |
|  | June 2022 | $0.99834 \pm 0.00007$ | −0.19834% | −0.0032 ppm | −0.00384 ppm (−0.213 %) |  |
|  | October 2022 | $0.99962 \pm 0.00002$ | 0.12525% | −0.0008 ppm | 0.00246 ppm (0.137 %) |  |
|  | March 2023 | $0.99872 \pm 0.00011$ | −0.09563% | −0.0023 ppm | −0.00154 ppm (−0.086 %) |  |

It is assumed that the reference in Karlsruhe does not drift in time. Therefore, a deviation before and after a campaign is due to a change of the TS. Hence, the presented difference gives an uncertainty to the final comparisons (compare with Appendix B and Figure 16).

**ILS-analysis:** A further monitoring tool is the measurement of the instrumental lineshape (ILS) of the TS. The ILS is described by two values, the modulation efficiency (ME) and the phase error (PE). The ME and PE are described in Hase et al. (1999).



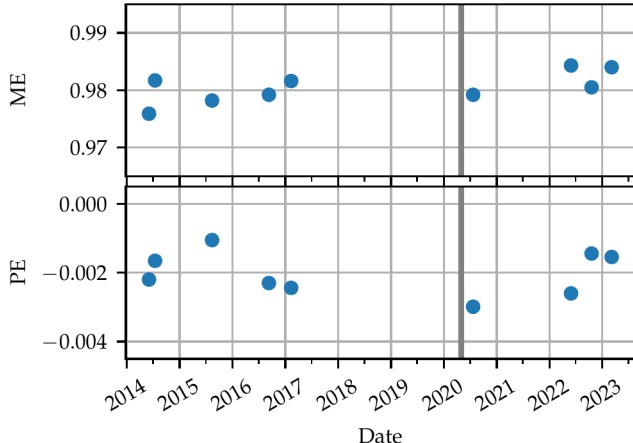

**Figure 3.** The ILS parameters of the spectrometer SN39 used as the TS measured at different dates. It is described by the Modulation Efficiency (ME) and the Phase Error (PE). The grey line indicates the date after which the measurements are relevant for this paper. The data before are plotted to show the values in the context of the history of the instrument.

In short, assuming a monochromatic wave, the ME describes the decrease of the envelope of the sinusoidal interferogram towards higher optical path differences (OPDs). The phase error describes the shift of the zero crossings of the sinusoidal interferogram. Both values describe the deviation of a real-world instrument to a theoretical instrument. For a theoretical perfect instrument one expects $ME = 1$ and $PE = 0$.

The ILS is measured before and after each visit. The results are plotted in Figure 3. The measurements collected before 2020
are not of relevance for the data evaluation of the TS, as always the newest available ILS value is used for the retrievals with PROFFAST2. However, they are listed in the figure to provide a comparison with the historical data of its ILS.

As a measure of the stability, the mean and the standard deviation of the ME and the PE are calculated over all measurements in Figure 3. For the ME this gives $0.98051 \pm 0.00272$, for the PE $-0.00202 \pm 0.00063$. As a comparison, the mean and the standard deviation for the ME and PE values of the reference instrument SN037 as published in Alberti et al. (2022a) are
$ME = 0.98361 \pm 0.00267$ and $PE = 0.00145 \pm 0.00122$. These values are in the same order of magnitude showing that the ME and PE of the TS instrument are within the normal range of an EM27/SUN spectrometer.

## 3.1 Variable bias in XCO

To find the reason for the variable differences of the XCO product several potential error sources have been investigated.

The first idea is that channeling might be responsible for the observed variations. Channeling describes the phenomenon of
a thin element in the optical path which acts as a cavity and resonantly amplifies a certain frequency or integer multiples of it (Blumenstock et al., 2021). This has already been demonstrated by Frey (2018). This problem was ameliorated for all new EM27/SUN spectrometers by adding an antireflection coating on the longpass filter. However, the instrument SN039 used as TS is the prototype version of the dual channel setup (see Hase et al. (2016)). In the laboratory, measurements to check for





channeling as described by Frey (2018) are collected. They seem to be free of channeling which does not support the thesis of
channeling being the source of the deviation.

Next, a misalignment of the optics of the second channel could contribute to the variable XCO bias. To investigate this, a second XCH4 product, called $\mathrm{XCH_4^{S5P}}$, which is retrieved from an alternative window within the range of the second channel is plotted in Figure 2. This product does not show the same behavior as the XCO retrieval does. This can be seen also in Figure 4 which shows the SZA dependency of $\mathrm{XCH_4}$ and $\mathrm{XCH_4^{S5P}}$. In addition the alignment of the second channel was checked
by opening the instrument. Also by this method, no misalignment could be detected. The excellent agreement of $\mathrm{XCH_4}$ and $\mathrm{XCH_4^{S5P}}$ also rules out an ILS problem or a zero baseline problem in the second channel.

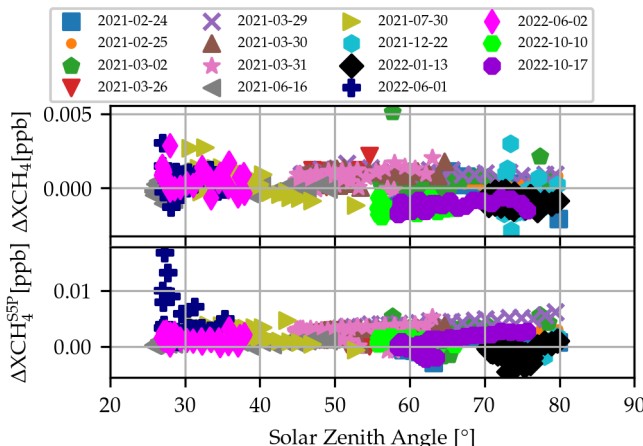

**Figure 4.** Investigating the dependency of $\mathrm{XCH_4}$ and $\mathrm{XCH_4^{S5P}}$ from the SZA. The data do not show a clear SZA dependence. This supports the thesis that there is no misalignment of the second channel causing the seasonal variability in XCO, because otherwise this would lead to the same dependency as given in Figure 5 for XCO.

Fortunately, a larger dataset of side-by-side measurements exists covering 15 measurement days starting from 2021-02-24 until 2022-10-17. This dataset supports the hypothesis of an XCO bias which depends on the SZA. This is visualized in Figure 5, where the $\Delta$XCO between the reference instrument and the TS instrument is plotted as a function of the solar zenith angle.
From this data it is possible to derive a empirical correction by fitting a linear regression line to the data. The result is the empirical correction function

$$c_{\mathrm{XCO}}(\mathrm{SZA}) = 7.39076 - 0.071271 \cdot \mathrm{SZA[deg]} \tag{3}$$

which is applied to all XCO data measured by the TS in this paper, except the data plotted in Figure 2 in orange to demonstrate the effect of the correction.



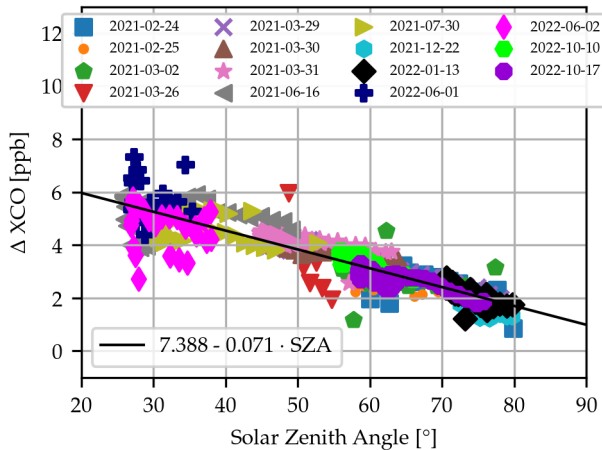

**Figure 5.** The $\Delta$XCO of the reference EM27/SUN spectrometer and the TS device as a function the solar zenith angle (SZA). There is a clearly visible dependence on the SZA. The reason for this is still under investigation. However, this dependence is used to derive an empirical linear correction of the XCO values. The correction is applied to all measured data in this paper. In Figure 2 the corrected XCO values are plotted using red "x"-shaped markers.

### 3.2 Verification of the Pressure Sensor Used In the Travel Standard

The TS is equipped with a Vaisala PTB330 pressure sensor acquired in April 2021. Part of the verification performed at KIT is to also compare the pressure measurements collected by the TS sensor with the pressure data used for the Karlsruhe TCCON retrieval. For the Karlsruhe TCCON station the pressure data of a nearby weather station (Rheinstetten, 15 km south-south-west of the TCCON station) of the German weather service (Deutscher Wetterdienst (DWD)) is used. Unfortunately, there was an unnoticed crash of the program used to collect the pressure data of the TS sensor before the Tsukuba and the Canada campaign such that there is no side-by-side data for those periods. The only measurements are available after the Canada and Australia campaign. (For the evaluation of the solar side-by-side measurements, the pressure data recorded by the "Rooftop sensor" (introduced below) is used.) They are plotted in Figure 6. The data show an excellent agreement with the height corrected data of the DWD-Rheinstetten station. The bias compensating factor between the TS and the DWD data is $K_{\mathrm{TS}}^{\mathrm{DWD}}(\mathrm{Can}) = 0.999813$ and $K_{\mathrm{TS}}^{\mathrm{DWD}}(\mathrm{Aus}) = 0.999924$ before and after the Australia campaign. The change of the bias compensation factors is $-0.1‰$. For further calculations the average of both is used, which is $K_{\mathrm{TS_P}}^{\mathrm{DWD_P}} = 0.999869$. For an average pressure of 1000 hPa this gives an average deviation of $0.131$ hPa. According to the datasheet of the sensor (Vaisala, 2023) the accuracy of the sensor is $0.1$ hPa. Therefore, the deviation to the DWD sensor is only slightly above the sensors accuracy which is an excellent agreement considering that the DWD station is 15 km away and the data are height corrected.

In Figure 6 we plotted the measurements from another Vaisala PTB330 sensor measuring at the rooftop terrace on the 7th floor of the institutes building. This sensor is called "Rooftop (RT) sensor" in the following. The agreement between the RT and the TS sensor also is excellent. The rooftop sensor collected data longer than a year, so we can use its data as a proxy



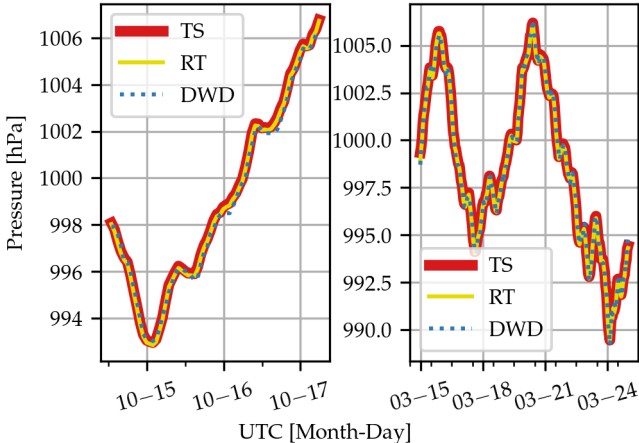

**Figure 6.** Measurements of the Vaisala PTB330 sensor in the TS compared with a weather station of the German weather service (DWD) in Rheinstetten and a second Vaisala PTB330 mounted permanently on the institutes rooftop (RT). The measurements in the left and right panel are collected in October 2022 (between Canada and Australia) and March 2023 (after Australia), respectively. For the comparison before and after Australia a bias compensation factor of $K_{\mathrm{TS}}^{\mathrm{DWD}}(\mathrm{Can}) = 0.999813$ and $K_{\mathrm{TS}}^{\mathrm{DWD}}(\mathrm{Aus}) = 0.999924$ is found, respectively. The data of the Rheinstetten DWD station are corrected for altitude difference of 17 meter.

to investigate the stability of the PTB330 sensors. The comparison is shown as a scatter plot in Figure 7. The data show an excellent agreement. A function $p_{\mathrm{RT}}(p_{\mathrm{DWD}}) = a \cdot p_{\mathrm{RT}}$ fitted to the data results in $a = 0.999984 \pm 3.061845 \cdot 10^{-6}$. The deviation averaged over the whole period is $0.0138\,\mathrm{hPa}$. This shows the high stability of the PTB330 sensors and hence it is justified to use it as a reference with the TS.

## 4 Description of the TCCON and Travel Standard Data Sets Collected in Tsukuba, Japan

In this section, we analyze the data recorded in Tsukuba, Japan. A quantitative comparison of the site-to-site bias is done in Section 7.2 together with the results of the other sites visited. The Tsukuba TCCON station is located at 31 meters above sea level (masl), the TS collected its measurements at an altitude of 39 masl. The TS was operated in Tsukuba from 2022-03-24 until 2022-04-25. In this period we collected 8 days of measurements. The low-resolution data measured with the Tsukuba TCCON instrument will be denoted as TK-LR (Tsukuba-low-resolution), the standard TCCON data as TK-HR (high-resolution) and the data of the Travel Standard as TS.

**Pressure analysis**: As aforementioned the TS is equipped with a Vaisala PTB330 pressure sensor. Unfortunately, during the first campaign of the TS in Tsukuba, the sensor was integrated into the enclosure such that the sensor was measuring the pressure within the enclosure. While analyzing the data after the campaign, we realized that the venting cooling fan in the enclosure produced a significant dynamic pressure inside the enclosure. As a consequence, the recorded pressure data were not





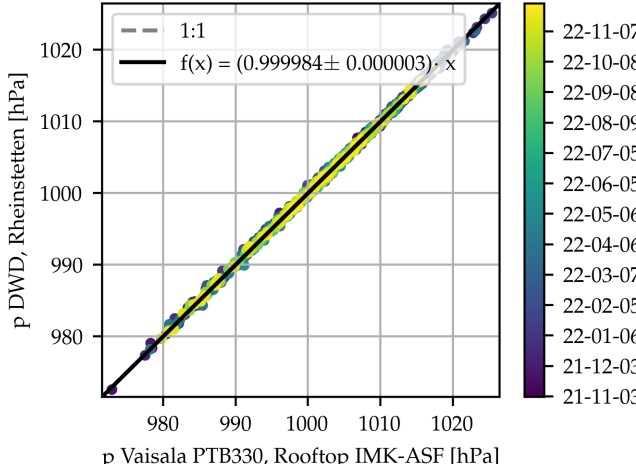

**Figure 7.** Results of the comparison of a Vaisala PTB330 mounted on the terrace of the institutes building at the 7th floor and the DWD weather station in Rheinstetten in 16 km distance of the institute. The scatter plot shows an almost perfect agreement. The data do not show a drift in time. A linear function $p_{\mathrm{RT}}(p_{\mathrm{DWD}}) = a \cdot p_{\mathrm{RT}}$ fitted to the data gives in $a = 0.999984 \pm 3.061845 \cdot 10^{-6}$. The deviation averaged over the whole period is $0.0138\,\mathrm{hPa}$ which is smaller than the accuracy of the PTB330 sensor which is $0.1\,\mathrm{hPa}$ (Vaisala, 2023). This shows the stability of the Vaisala PTB330 sensors.

usable for the retrieval. For future campaigns a tube was used which is connected to the inlet of the pressure sensor and ends at the outside of the enclosure to sample the surface pressure outside of the enclosure.

Fortunately, a side-by-side measurement with the pressure sensor of the Tsukuba TCCON site was recorded with the fan turned off. Using this subset of data, it was possible to calculate a factor to map the data recorded with the pressure sensor of the Tsukuba TCCON site to the pressure sensor of the TS. Hence, for the retrieval of the TS and the TK-LR pressure data of the official TCCON evaluation are used but with a correction for the altitude and a second factor to match it to the level of the TS pressure Sensor.

The pressure side-by-side measurements are plotted in Figure D1. They were recorded from 2022-04-23 until 2022-04-24 each at midnight local time. Both datasets are resampled to one minute bins. The Tsukuba pressure record is slightly lower than the TS record by $-0.105\,\mathrm{hPa}$, causing an bias compensation factor of $K_{\mathrm{TK_p}}^{\mathrm{TS_P}} = 1.00010$. The pressure offset is small enough that we do not expect it do influence the XGas retrieval.

**XAIR analysis:** In Figure 8 the retrieved data of XAIR, $XCO_2$, $XCH_4$ and XCO are plotted. The TS data are in blue, the
TK-LR data are in orange and the TK-HR data are in green.

The TK-LR and TK-HR XAIR data show a clear airmass dependency over the course of the day. This is an indicator for an error in the recorded timestamp of the interferograms, which leads to a wrong calculation of the solar position. To correct this erroneous timestamp, empirically a correction of $-44\,\mathrm{s}$ is found for the TK-LR data. The resulting data are plotted in the





attachment in Figure C1. It can be seen clearly that the airmass dependency of XAIR is almost completely eliminated by this.
Furthermore, this also influences the XGas retrievals but to a much lesser extent. This is because in first order the the timing error cancels out when calculating XGas (compare with Equation (4)). The reason for this time offset is still under investigation and therefore no time-corrected TK-HR data are available yet. Note, that the TK-HR data shown here is not the official TCCON product as the time error will be corrected before submitting the data to the TCCON database. TCCON is routinely doing a QA/QC check before publishing data, which is expected to discover such an error. However, this error was discovered first by
the TS campaign data analysis.

Further analysis is conducted for both, the corrected and uncorrected TK-LR data as well as for the uncorrected TK-HR data. The the corrected data will be denoted as TK-LR-tcor the uncorrected as TK-LR.

The XAIR values of the TS are normally distributed around unity. The only exception is the 8[th] of April 2022 where XAIR recorded by the TS oscillates during the morning hours. These oscillations seem to be induced by the pressure record, which
shows the same oscillations as well. These oscillations are also detected by the pressure station of the Japan Meteorological Agency in Tsukuba (Tateno) (Japan Meteorological Agency, 2023). These quasi-periodic pressure variations might be an effect of mountain wave activity generated by the surrounding summits. The wave activity in this area can be extreme as the loss of flight BOAC911 teaches (Dempsey, 2023).

Normally, one does not expect that a change in the surface pressure is influencing the XAIR retrieval. The reason why in
this case the pressure variations can be seen in XAIR is the following: For the calculation of XAIR a hydrostatic atmosphere in equilibrium is assumed. However, in presence of those waves hydrostatic equilibrium can no longer be assumed and hence, they directly disturb XAIR.

Fortunately, the oscillation of the pressure ends before the TCCON-measurements are started, hence it does not influence the side-by-side evaluation.
XAIR is designed such to scatter around unity for a well aligned and set up instrument. Its distribution around unity is measured by calculating the mean value and the standard deviation of XAIR. For the TS this is $0.99796 \pm 0.00091$, $1.00224 \pm 0.00482$ for the TK-HR, and $0.99778 \pm 0.00357$ for the TK-LR data. For the TK-LR-tcor data it becomes $1.00028 \pm 0.00184$. The values show clearly that the time-correction improves the XAIR data significantly for the TK-LR data.

**XGas analysis:** For both the TK-HR and the TK-LR data one can see high noise levels for XAIR, $XCO_2$ and $XCH_4$. This is
due to a pronounced intensity drop for large wavenumbers in the Tsukuba TCCON spectra and is discussed in detail in Section 4.1.

For $XCO_2$ and $XCH_4$ a good agreement is found for the TS data and the TCCON data. Taking the average over all days and subtracting the TK data from the TS data gives an average bias over all days for $XCO_2$ of $-0.0209$ ppm and $0.2661$ ppm for the low- and high-res data, respectively. For the TK-LR-tcor data the bias is $-0.0267$ ppm. For $XCH_4$, we find a bias of
$0.0028$ ppm and $-0.0046$ ppm for the low and high-res data and $0.0027$ ppm for the TK-LR-tcor data. For XCO the overall mean is $-1.5997$ ppb for the TK-LR data and $-1.5768$ ppb for the TK-LR-tcor data. In contrast, for the TK-HR data there are days with better agreement and others with worse agreement, resulting in an overall mean bias of $-8.7191$ ppb. To check if this is a problem with the PROFFAST retrieval software, the TK-LR data are also processed using GGG2020, plotted with





yellow "x"-shaped markers. The day to day variability is similar to the TK-LR data processed with PROFFAST, even though
the overall mean difference is $3.02\,\mathrm{ppb}$ larger. This indicates that it is not due to an issue with the PROFFAST code.

We therefore assume that the origin of the high day-to-day difference is due to a known issue with the CO a prioris shared
by both analysis softwares, GGG and PROFFAST. The GEOS FP-IT model used for generating the priors incorporates an
outdated emission inventory. This causes an overestimation of the CO a prioris in urban or energy-intensive areas. The resulting
unrealistic CO a-priori profile in combination with the different column sensitivities (due to the different spectral resolutions)
causes the observed bias in the XCO data (Laughner et al., 2023a; Laughner, 2023).

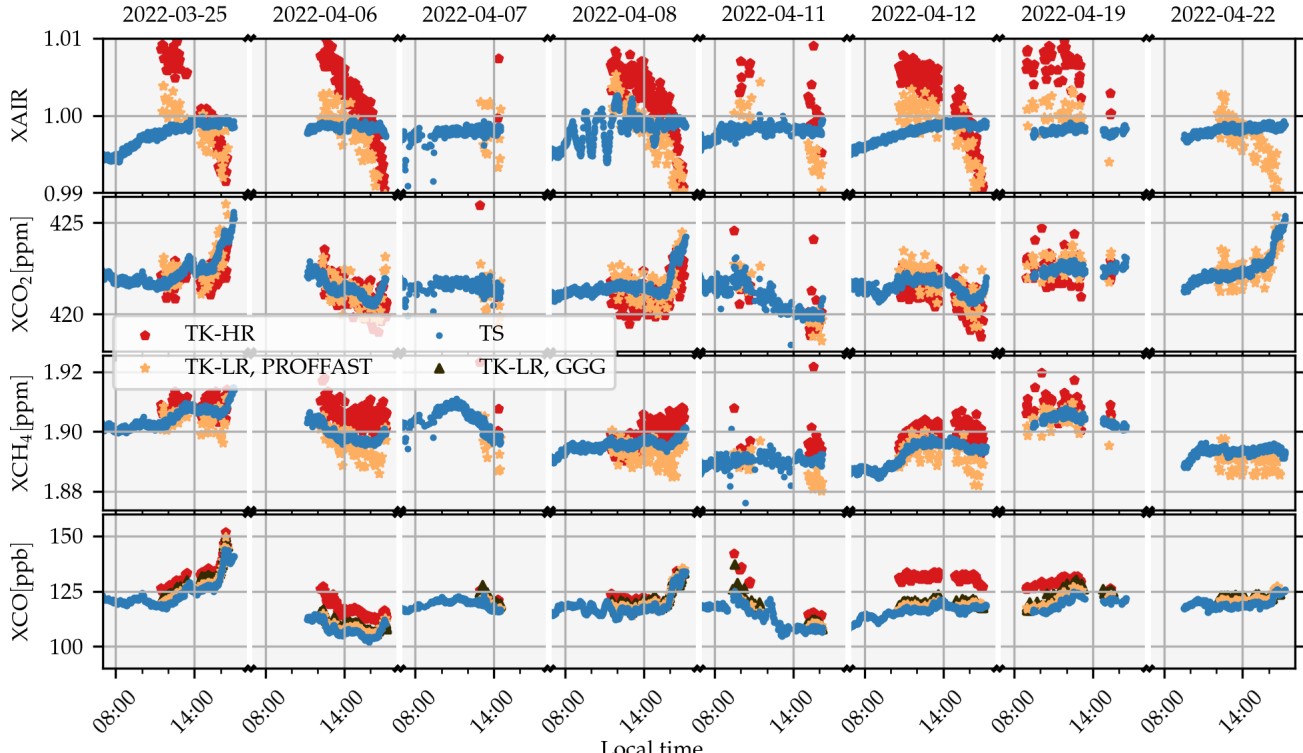

**Figure 8.** The XAIR and XGas results for $XCO_2$, $XCH_4$ and XCO of the side-by-side measurements in Tsukuba, Japan. In blue the results
of the retrieval of the TS, in orange of the low-resolution data (both processed with PROFFAST2) and in green the XGas values retrieved by
the high-resolution TCCON spectra (processed with GGG2020) are plotted. One can see a good overall agreement for $XCO_2$ and $XCH_4$.
For XCO the agreement between the TK-HR and the TS data varies from day to day. This is caused by a combination of unrealistic a-priori
and different spectral resolutions. Furthermore, the TCCON results are noisier than the TS results. The origin of this is a signal drop for
higher wavenumbers in the spectrum. The fast oscillation of XAIR in the morning of 2022-04-08 are due to presumably non-hydrostatic
pressure oscillations measured independently by the weather station of the Japan Meteorological Agency in Tsukuba, too.





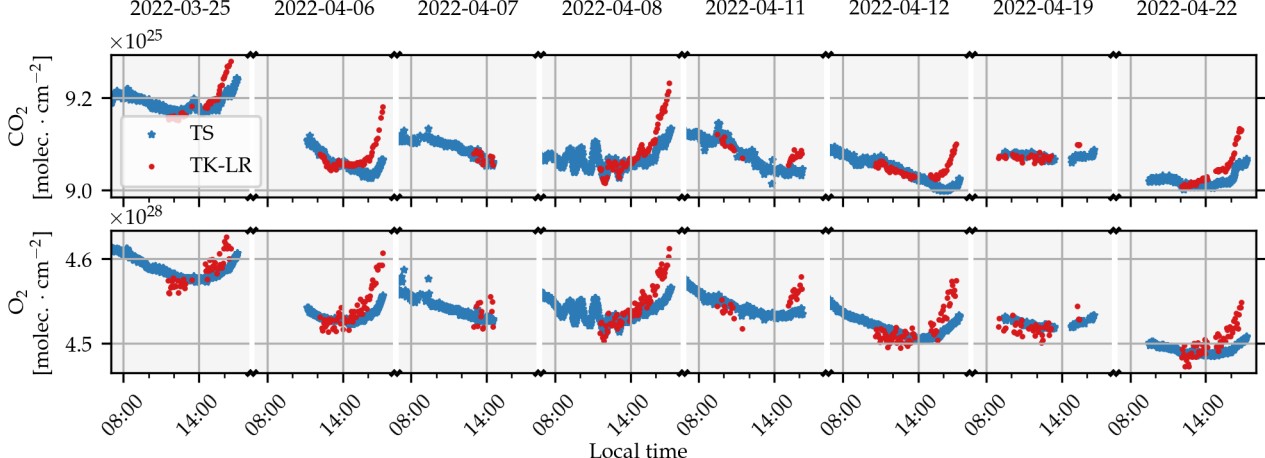

**Figure 9.** Retrieved total column values of $CO_2$ and $O_2$ for the TS data in blue the TCCON-low-resolution data in orange. For the TCCON-lr data the noise in the $O_2$ retrieval is much higher than it is for $CO_2$. This results in noisy XGas values which are shown in Figure 8. The reason for this is in a low signal level of the TCCON-LR spectra, as depicted in Figure 10.

### 4.1 Investigation of high noise levels in TCCON XGas values

Despite the good mean agreement, the TCCON-LR XGas values have a noticeably higher noise than the TS values. The reason for this has been found in the retrieval of the $O_2$ column. All XGas values are calculated by

$$XGas = \frac{VC_{\text{gas}}}{VC_{O_2}} \cdot 0.2095. \tag{4}$$

Here, $VC_{O_2}$ is the vertical column number of molecules per square centimeter of $O_2$ and $VC_{\text{gas}}$ the vertical column amount of the corresponding gas. Hence, a high scattering in the $O_2$ column influences all other XGas values. This is depicted in Figure 9, where the vertical column values of $O_2$ and $CO_2$ are plotted. The reason for the high scattering in the $O_2$ retrieval was found in the shape of the spectra recorded by the TCCON spectrometer. It is shown in Figure 10 in green. The maximum of the spectrum is normalized to unity. For illustration a spectrum of the Karlsruhe TCCON station is plotted in dark blue.

(Since the Karlsruhe TCCON setup differs from the standard setup used in the TCCON, the Karlsruhe spectrum drops to zero at $5450\,\text{cm}^{-1}$. It is normalized such that its maximum matches with the spectrum height of the Tsukuba spectrum at the same wavenumber.) To characterize the observation in Tsukuba, two values were calculated: The first is the maximum value, $\max_{O_2}$, within the $O_2$ window. The second value is the signal-to-noise ratio ($SNR_{O_2}$) in the $O_2$ window. This was calculated by taking standard deviation $\sigma$ of the parts of the spectra without signal (i.e. at the upper and lower end of the spectra or of points of zero

transmittance). The signal-to-noise ratio is then calculated via $SNR_{O_2} = \frac{\max_{O_2}}{\sigma}$. The spectra are normalized to unity before doing this calculation.

As a consequence of this finding, the spectra of the TCCON visited with the TS plus several additional TCCON stations were checked. The results are summed up in Table 2. The results vary significantly across the sites. From this table we expect





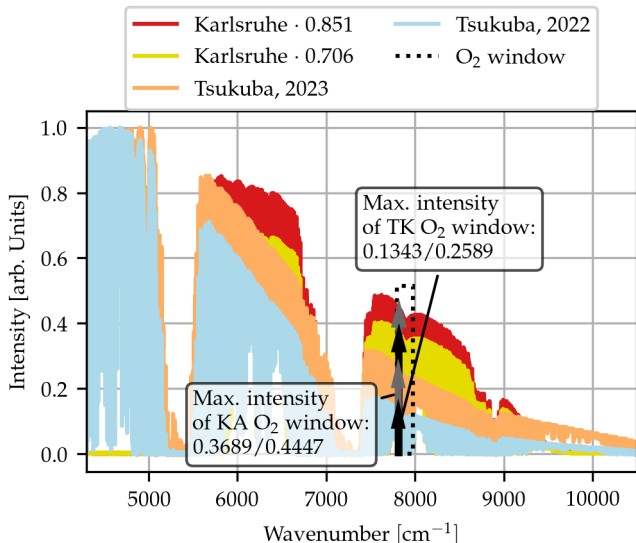

**Figure 10.** Comparison of the Tsukuba and the Karlsruhe spectra. The Tsukuba spectrometer has been re-aligned in early 2023. Therefore, a spectrum recorded during the TS visit in 2022 is plotted in green and the re-aligned spectrum is plotted in orange. For comparison, the Karlsruhe spectrum is plotted in dark and light blue. The Tsukuba spectra are normed to unity, the Karlsruhe spectra are normed to match the intensity of the Tsukuba spectrum at $5680 \mathrm{cm}^{-1}$. The Karlsruhe spectrum in light blue is normed to the 2023 Tsukuba spectrum and the spectrum in dark blue is normed to match with the 2022 Tsukuba spectrum. The reason for the Karlsruhe spectrum drops to zero at $5450$ $\mathrm{cm}^{-1}$ is the non-standard TCCON setup in Karlsruhe. The Tsukuba spectra decrease strongly towards higher wavenumbers. However, after the realignment in 2023 the decrease is less intense. For the $O_2$ retrieval the low signal level at the spectral position of the 1.26 μm $O_2$ band results in a bad signal-to-noise ratio and hence noisier XGas data. As a quantifying metric for assessing the spectrum, the maximum in the $O_2$ window is determined.

high scatter also for the Wollongong station, which is confirmed by the later analysis (see section 6). It is interesting to see that
for sites which set up a new instrument recently, the values are much better for the new instruments.

From the instrumental view point, this signal drop is likely created by the characteristics of the beam splitter and of the detector element. Also mirror degradation and deterioration of other optical elements might have an influence on this. Furthermore, it is influenced by the alignment of the spectrometer: In early 2023 the Tsukuba spectrometer was realigned. This causes the intensity drop to be less pronounced. The realigned spectrum is plotted in Figure 10 in orange and in light blue the
Karlsruhe spectrum as comparison. In Table 2, the values for Tsukuba in 2022 and 2023 before and after the realignment are given.



**Table 2.** Analysis of various spectra recorded at different TCCON sites. The spectra were all recorded at around noon on a bright day. In order to make $\max_{O_2}$ comparable, they are all normed to unity. The value $\max_{O_2}$ is the maximum value in the $O_2$ window, ranging from (7800 - 7980) $cm^{-1}$. The noise is described by the standard deviation of the parts without signal. The signal to noise ratio is calculated by dividing the max. value in the $O_2$ window by the calculated noise. One can see that there are large differences across the network.

| Site | $\max_{O_2}$ | signal-to-noise for $O_2$ |
|---|---|---|
| Rikubetsu | 0.3075 | 271.2298 |
| Burgos | 0.3395 | 392.7592 |
| Wollongong New | 0.5692 | 274.7110 |
| Wollongong Old | 0.1510 | 49.8372 |
| Karlsruhe | 0.5212 | 901.7539 |
| Tsukuba, 2022 | 0.1343 | 95.6686 |
| Tsukuba, 2023 | 0.2589 | 220.1736 |
| ETL | 0.2881 | 197.2654 |

## 5  Data Analysis of ETL, Canada

One day before the TS arrived at the East-Trout-Lake (ETL) TCCON site in Canada, the reference laser of the TCCON spectrometer broke down. Consequently, it was not possible to perform the planned side-by-side measurements. Hence, there is unfortunately no direct comparison of station XGas measurements with the TS.

**Pressure analysis:** It was possible to record side-by-side pressure data in the range from 2022-08-16 at 8:00 until 2022-08-17 at 20:00 local time. The data of are plotted in Figure D2. The ETL data are recorded every second. The raw data have a high noise level, however, for the retrieval an average is calculated. For the comparison, both the TS and the ETL data are resampled in 1-minute bins giving a good overall agreement. On average the ETL pressure records are $0.00386$ hPa larger than the TS pressure records. This results in an bias compensation factor of $K_{TS_p}^{ETL_p} = 0.9999959$.

**XAIR analysis:** The ETL TCCON spectrometer recorded 7 days of alternating high- and low-resolution measurements performed before the TS arrived. Furthermore the TS recorded three days of data when arriving there. The data are plotted in Figure 11. Still the data can be used to check for the noise level as well as to check for any anomalies in XAIR. The visual analysis does not reveal any anomalies. As for the Tsukuba data the mean and the standard-deviation of XAIR is calculated. For the ETL-HR data this gives $1.00043 \pm 0.00131$, for the ETL-LR data $0.99976 \pm 0.00163$ and for the TS data $1.00095 \pm 0.00082$. The data are all close to unity with little noise. Hence no instrumental problems are expected from this.

Furthermore, the data are used to check for the noise level. For this the XAIR and the XGas values of the ETL-LR and ETL-HR data are analyzed. From a visual inspection, it is already apparent that the noise level is lower than it is for the Tsukuba data. A quantitative analysis is provided in section 7.2.





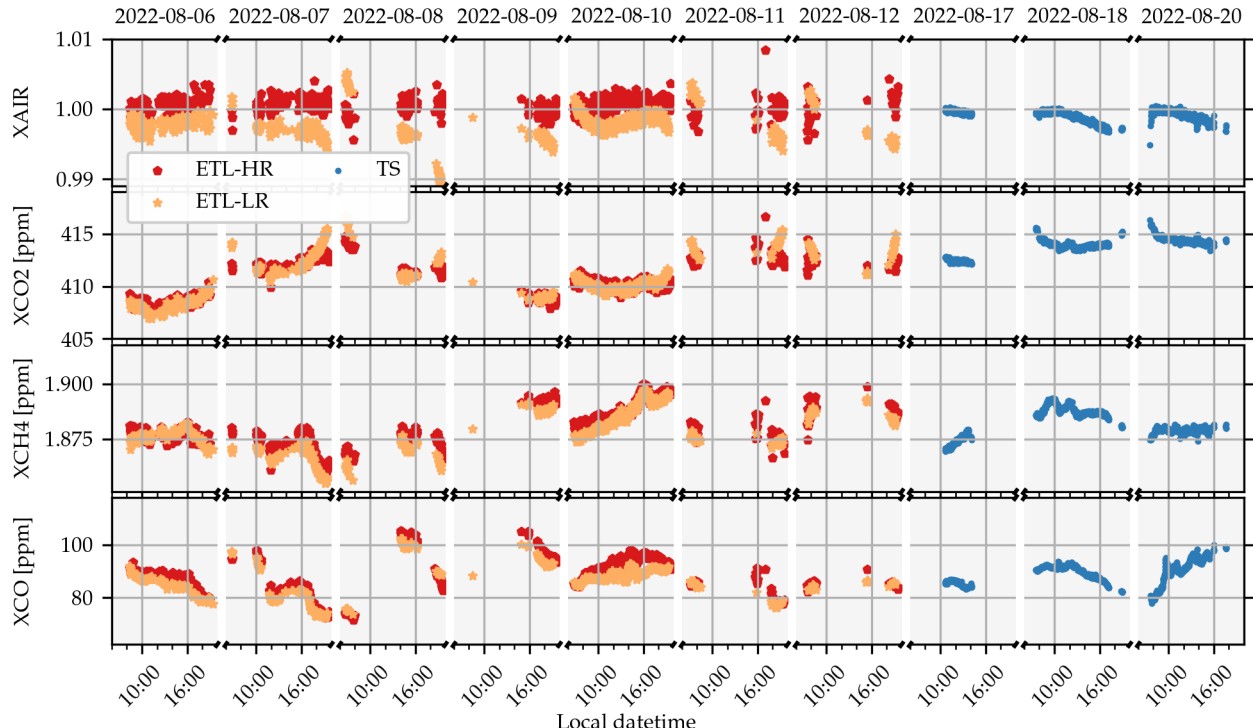

**Figure 11.** The retrieved XAIR and XGas values for the high- and low-res data in East Trout Lake (ETL), Canada. The data were recorded before the TS arrived. The reference laser of the ETL-TCCON spectrometer broke down when the TS was en route. Hence, no side-by-side measurements were possible. Nevertheless, the data are used for an XAIR and noise analysis.

## 6 Description of the TCCON and the TS Data Sets Collected in Wollongong, Australia

The TS visited Wollongong from 20222-12-06 until 2023-01-26. In this period 15 days of side-by-side measurements could be collected.

In Wollongong there are currently two TCCON stations, an old and a new one. The new one is not yet measuring continuously, hence, there are less data. The analysis in this work is therefore limited to the old instrument. The old instrument is located at 34.406 south, 150.879 east at an altitude of 35 masl. The new instrument is located at 34.406 south, 150.880 east at an altitude of 49 masl. The TS was placed at the rooftop next to the tracker of the new instrument but at an altitude of 48 masl.

**Pressure analysis:** The pressure sensors of the old and new TCCON site are at an altitude of 30 masl and 44 masl and the TS sensor at 48 masl. Hence, to compare the data, the records of the TS sensor are corrected for a height difference of $-4$ m and $-18$ m, using the barometric height formula with a temperature of $T = 22°C$ and the earth acceleration of $g = 9.81\ \frac{\mathrm{m}}{\mathrm{s}^2}$. The new TCCON and the corrected TS data are in good agreement with a small high bias of the TCCON data of $0.02517\ \mathrm{hPa}$. The old TCCON and the corrected TS data agree with a small low bias of the TCCON data of $0.02517\ \mathrm{hPa}$. This gives a bias compensation factor of $K_{\mathrm{WG_p}}^{\mathrm{TS}} = 1.0000373$.



In Figure D3 the pressure data collected during two days within this period are plotted. The days are chosen randomly. However, the analysis takes into account the whole dataset recorded during the visit.

Note that at the time this manuscript is written the altitudes of the TCCON pressure sensors and trackers remain with an uncertainty of around 1 m. The reason for this is that due to the visit of the TS an error in the so-far assumed altitudes of the pressure sensors and trackers was detected. The altitude of the tracker and the pressure sensors of the old TCCON site have been assumed to be both at 30 masl. The altitude of the new tracker and pressure sensors have been assumed to be at 34 masl and 30 masl. The new heights used here are determined using the pressure sensor of a smartphone.

The detection of this error is very important as the wrong height influences the retrieved XGas values. Furthermore, for the evaluation of the old TCCON data, the height difference of 5 m was not taken into account so far. This height difference leads to an approximate pressure difference of 0.58 hPa which is significant for the retrieval. As the GGG2020 dataset was not published at this time, the correction still could be included. For the GGG2014 dataset, however, this correction was not applied.

**XAIR analysis:** For the processing of the WG-LR and WG-HR data, the pressure data collected by the sensor at the old TCCON site with a height correction of 5 m is used. The data are plotted in Figure 12. The WG-HR data are shown in green, WG-LR data in orange and the TS data in blue. A visual analysis shows a good agreement of the XAIR values for all three measurement products. The mean and the standard deviation of XAIR is $0.99957 \pm 0.00253$ for the WG-HR data, $0.99881 \pm 0.00072$ for the WG-LR data and $0.99885 \pm 0.00023$ for the TS data. The high standard deviation of the TCCON 515 data is due to the high noise level.

**XGas analysis:** In the following the side-by-side measurements of the XGas values are discussed. Unfortunately, the WG-LR data were recorded with a low frequency, such that the timely distance between the measurements is in the order of 15 to 25 minutes. Hence, the bin size was chosen to be 30 minutes for the WG-LR data, instead of 10 minutes as chosen for the Tsukuba measurement. The low data amount makes it difficult to derive reliable statistical values. Consequently, the results of 520 the WG-LR data analysis might be less significant.

For $XCO_2$, $XCH_4$ and XCO the WG-HR data show a high noise level, too. Interestingly, the noise level of the WG-LR data is less than it is for the HR data. The reason for this is a higher signal-to-noise ratio in the WG-LR spectra compared to the WG-HR spectra. This can be seen clearly in Figure 13. This is discussed in more detail in Section 7.1.

The overall agreement is good for all gases. For $XCO_2$, averaging over all days, the differences of the TS minus the WG 525 data are 0.1316 ppm and 0.1374 ppm for the WG-LR and WG-HR data, respectively. For $XCH_4$ the mean differences are 0.0005 ppm and $-0.0025$ ppm for the LR and HR data. For XCO the mean differences are 3.1902 ppb and $-1.2482$ ppb for the LR and HR data.

Interestingly, for XCO the day-to-day differences of the HR and LR data are not as high as for the Tsukuba LR and HR data. This is probably because Wollongong is located in a more rural area than Tsukuba, and hence the CO priors are more realistic 530 (compare with the end of Section 4).





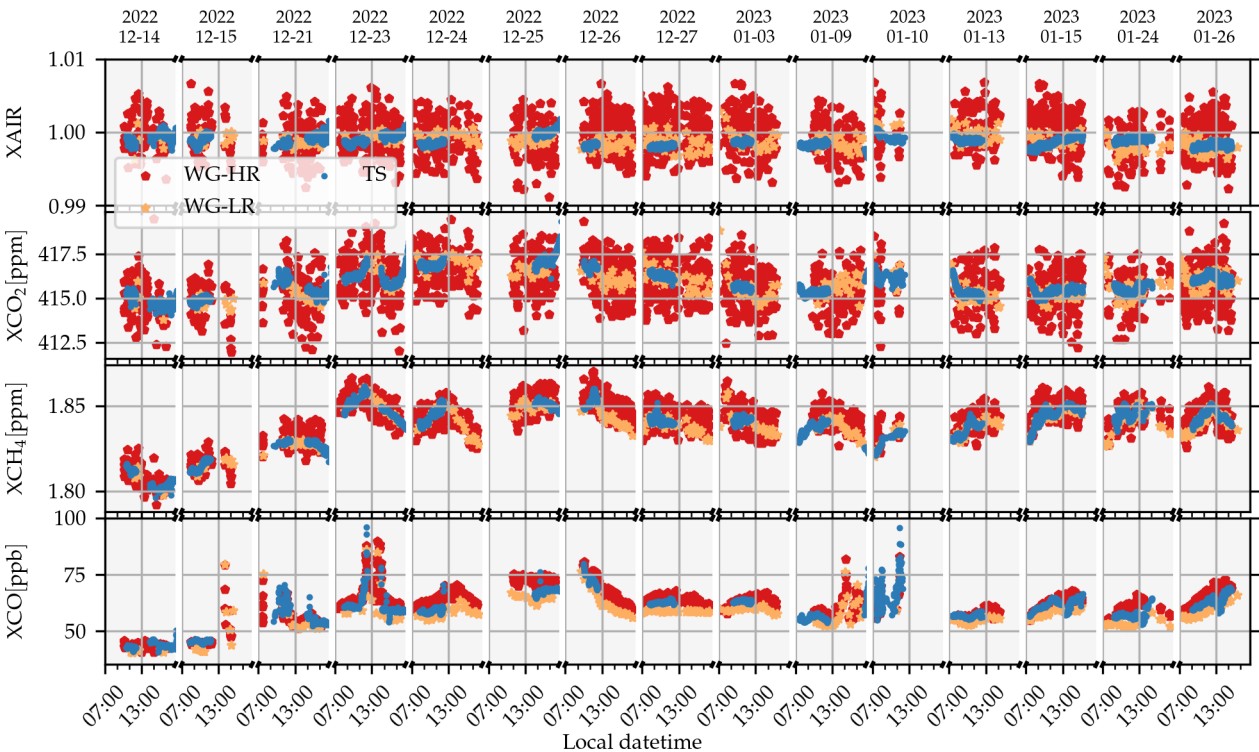

**Figure 12.** Raw data of the Australia Campaign. For all species the overall agreement is good. It is interesting to see that the HR data are much noisier than the LR data. This is discussed in section 7.1. Compared to the Tsukuba data, the difference of the XCO LR and HR data is smaller. This is probably due to better a priori profiles of the less urban area of Wollongong compared with Tsukuba.

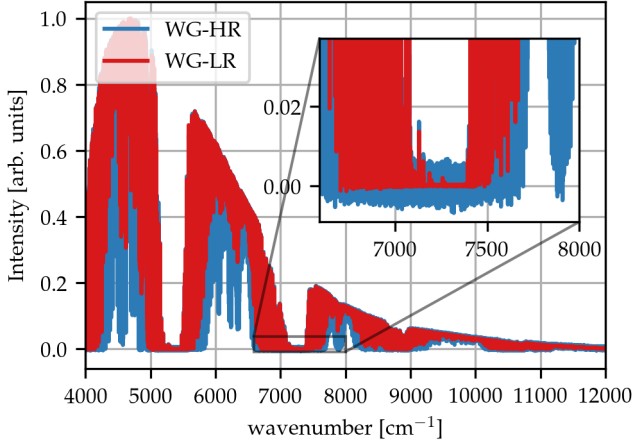

**Figure 13.** Comparison of the LR and the HR data of the TCCON spectrometer in Wollongong. The HR data show a significant worse signal to noise ratio than the LR data. This is clearly visible at the inset axes.



**Table 3.** Standard deviations $\sigma$ of the XGAS and XAIR values of the low- and high-resolution data of the visited TCCON sites. For all sites the low-resolution data are less noisy than the high-resolution data, except for XCO of Wollongong. The data are visualized in Figure 14.

| Species | TK-LR | TK-HR | ETL-LR | ETL-HR | WG-LR | WG-HR |
|---------|-------|-------|--------|--------|-------|-------|
| XAIR | 0.00145 | 0.00172 | 0.00054 | 0.00077 | 0.00072 | 0.00239 |
| $XCO_2$ [ppm] | 0.57111 | 0.64637 | 0.27585 | 0.28125 | 0.35094 | 1.07527 |
| $XCH_4$ [ppm] | 0.00279 | 0.00300 | 0.00108 | 0.00149 | 0.00235 | 0.00495 |
| XCO [ppb] | 1.21170 | 1.42212 | 0.44576 | 0.58671 | 2.22172 | 1.92952 |

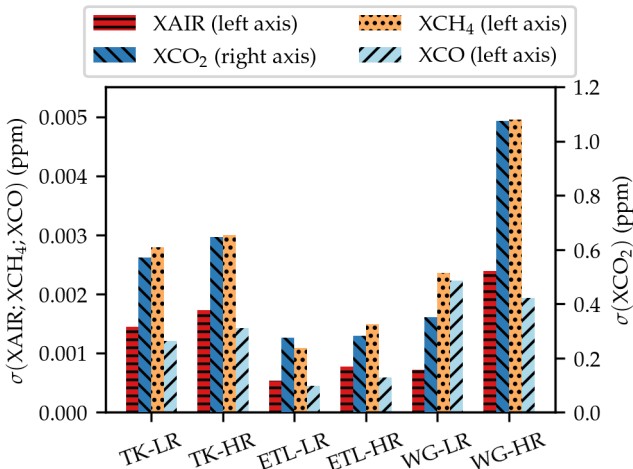

**Figure 14.** Visualization of the standard deviations as a measure the noise in the XGas time series of the sites visited with the TS. The data of the plot are also given in Table 3. This shows clearly, the different performance with respect to noise of the different TCCON spectrometer. Note that $XCO_2$ refers to the y-axis on the right and all other gases refer to the y-axis on the left. XCO is plotted in ppm to increase its visibility.

# 7 Quantitative Analysis of the Data

## 7.1 Quantitative Noise Analysis

The reason for a higher noise level of the Tsukuba data is shown in Section 4.1. To make a quantitative analysis, the standard deviation of the time series of all TCCON products shown in the Figures 8, 12 and 11 are calculated. This is done by calculating

a rolling mean of data points which are temporally spaced less than 20 minutes and then calculate the standard deviation of the difference of the smoothed and the original data. This method is used to remove trends in the data.

The results are summed up in Table 3 and visualized in Figure 14. The different noise levels which can be estimated already from the time series plots of the data are also confirmed quantitatively. One can see that for all gases, except for XCO in Wollongong, the noise level for the LR data is lower than for the HR data. This is reasonable for two reasons which are at





an interplay here: On the one hand side, the spectral noise in an FTIR measurement increases steeply with maximum optical path difference (Davis et al., 2001). At the other hand side a lower resolved spectrum is not resolving the spectral absorption lines as clear as an higher resolved spectrum. Hence, strong absorbers like $CO_2$ or $CH_4$ are well resolved even with a low-resolution spectrometer. Hence they can profit from the higher spectra signal-to-noise ratio of a low-resolution spectrum. In contrast, weak absorber, like CO are not as good resolved in a low-resolution spectrum and hence are often better retrieved

from high-resolution spectra.

### 7.2 Derivation of the XGas Station to Station Bias

In this section the TCCON sites are quantitatively compared relative to the Karlsruhe TCCON site. The choice to use Karlsruhe as a reference was made since the COCCON reference device is regularly tied to the Karlsruhe TCCON station. This does not imply that the Karlsruhe TCCON serves as an absolute reference to the whole TCCON network. But the use as reference for

relative comparisons is an obvious choice.

Technically, the comparison is made by the usage of gas-specific bias compensation factors. They are determined as described in Appendix A. In the following it is assumed that the bias compensation factors fully describe the systematical bias between two spectrometers. Hence, in this ideal assumption we can write,

$$\overline{\mathrm{XGas_{XX}}} = \overline{\mathrm{XGas_{YY}}} \cdot K_{\mathrm{YY}}^{\mathrm{XX}} . \tag{5}$$

with $\overline{\mathrm{XGas}}_{\mathrm{XX/YY}}$ is the temporal mean of device XX or YY respectively. This allows us to retrieve a 'virtual' bias compensation factor to compare the TCCON site visited with the TS to the Karlsruhe-TCCON site. This is done by the multiplication of the bias compensation factors retrieved before each campaign in Karlsruhe (see Section 3) with the factors retrieved during the campaigns (given in Table A1.) This scheme is depicted in Figure 15 and described in Appendix B1. The resulting correction factors are given in Table A2.

To derive an more intuitive comparison, the bias compensation factors comparing the visited TCCON sites to the reference in Karlsruhe are used to calculate deviations in percentage. The calculations for this are given in Appendix B1. The resulting values are given in Table 4.

To assess the quality of the comparison it is crucial to do an error analysis for these 'virtual' bias compensation factors. For this two different contribution factors are considered: The first is the random error originating from the individual bias

compensation factors as described in Appendix A. The random error is given with a "$\pm$" sign. The second is an uncertainty introduced by a potential drift of the TS instrument relative to the COCCON reference. Here, the upper limit of this uncertainty is estimated by using the $\Delta K_{\mathrm{SN39}}^{\mathrm{SN37}}$ of the bias compensation factors measured before and after each campaign as given in Table 1. The details of the error calculation are carried out in Appendix B.

The error calculation was conducted for the bias compensation factors in Table A1 as well as for the deviations in percentage

in Table 4.





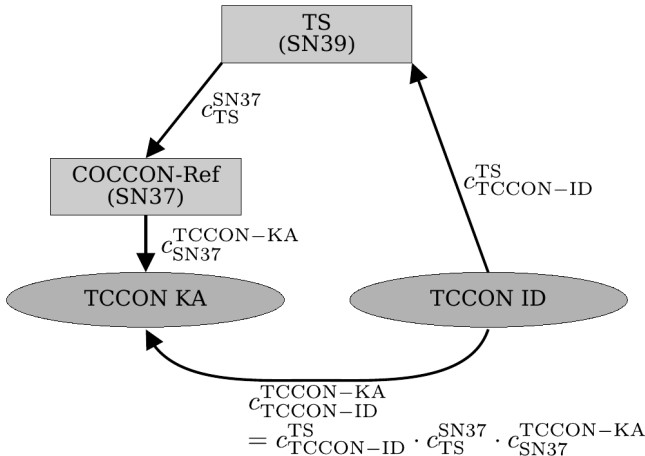

**Figure 15.** A graphical representation on how to use the correction factors to compare the measurements of a visited TCCON site to the Karlsruhe TCCON site.

**Table 4.** The table gives the deviations in percentage of the visited TCCON sites to the reference in Karlsruhe. The first error given is the random error emerging from the noise of the measurements. Second, a calibration uncertainty is given, which is calculated by considering a potential drift of the TS device relative to the COCCON reference device (derived from the $\Delta(\Delta\overline{\mathrm{XGas}})$ in Table 1).

| Site | Species | $\Delta^{\mathrm{SN37}}_{\mathrm{XX\text{-}LR}}$ [%] | $\Delta^{\mathrm{SN37}}_{\mathrm{XX\text{-}HR}}$ [%] |
|---|---|---|---|
| TK | $XCO_2$ | $0.11289 \pm 0.00826 - 0.06314$ | $0.02760 \pm 0.00839 - 0.06309$ |
| | $XCH_4$ | $-0.18871 \pm 0.00869 + 0.06685$ | $0.19398 \pm 0.00906 + 0.06711$ |
| | $XCO$ | $1.18157 \pm 0.04809 + 0.05455$ | $7.11865 \pm 0.04916 + 0.05775$ |
| TK | $XCO_2$ | $0.16401 \pm 0.00830 - 0.06318$ | $-$ |
| t-corr | $XCH_4$ | $-0.1115 \pm 0.00873 + 0.06690$ | $-$ |
| $-44\,\mathrm{s}$ | $XCO$ | $1.46537 \pm 0.0487 + 0.05470$ | $-$ |
| WG | $XCO_2$ | $0.01264 \pm 0.00744 - 0.07104$ | $0.00163 \pm 0.01023 - 0.07103$ |
| | $XCH_4$ | $-0.09253 \pm 0.00840 + 0.07089$ | $0.06115 \pm 0.00956 + 0.07100$ |
| | $XCO$ | $-5.57937 \pm 0.23080 + 0.55486$ | $1.82105 \pm 0.11168 + 0.59835$ |





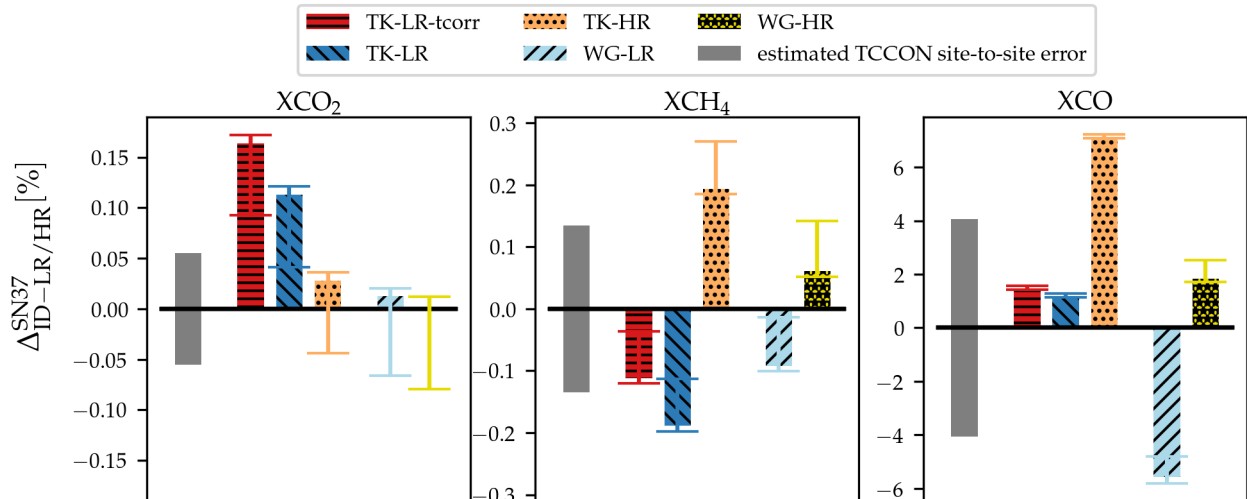

**Figure 16.** Results of the campaigns in Tsukuba, Japan and Wollongong, Australia. The three panels show the results for each species. In grey, the TCCON error budget as estimated in Laughner et al. (2023b) (Table 3, column "Mean abs. dev.") is plotted. The data of this plot are also given in Table 4. For a discussion of the results see in the main text.

### 7.3 Discussion of the Quantitative TS vs TCCON Comparison

The direct comparison of the visited TCCON sites to the reference in Karlsruhe TCCON site as a deviation in percentage is given in Table 4. In Figure 16 the data results are visualized. For the creation of Figure 16 we assume that the TCCON site-to-site error budget is distributed evenly around the TCCON-KA level.

The error bars are dominated by the calibration error introduced by the comparison of the TS unit SN39 with the COCCON reference unit SN37.

For $XCO_2$, when assuming an evenly distributed TCCON site-to-site error budget around the TCCON-KA level, as shown in Figure 16, the deviation of the Tsukuba-LR data are outside of the error budget, all other are within the budget. It is interesting to see that the correction of the timing error of the TK-LR data makes the deviations larger. Therefore, it is interesting to redo

this analysis with the corrected TK-HR data as soon as they are available.

For $XCH_4$ the low-, and high-resolution data of both, the Tsukuba and the Wollongong data deviate in the opposite directions. Future employments of the TS will tell whether this is a general feature of $XCH_4$. The WG data are within the TCCON site-to-site deviation budget, whereas this is not the case for the TK data, assuming again the TCCON deviation-budget is evenly distributed. For TK, the time-correction decreases the deviation.

For XCO and a centered deviation-budget, the deviations for TK-HR and WG-HR are larger than the budget, the rest is within the budget. The reasons for that are the following. First the already discussed deviation of the TK-HR and the TK-TS data is visible clearly, which is caused by the unrealistic CO a-priori profile. In contrast, the TK-LR results are almost within the error budget.



**Table 5.** Deviation of the pressure data recorded at the TCCON-site to the pressure sensor included to the TS and to a measurement station of the German Weather Service (DWD). The deviation in hPa is calculated by assuming a pressure of 1000 hPa.

| Site (XX) | $k_{\mathrm{ID_P}}^{\mathrm{TS_P}}$ | $k_{\mathrm{TS_P}}^{\mathrm{DWD_P}}$ | $k_{\mathrm{ID_P}}^{\mathrm{DWD_P}}$ | $\Delta_{\mathrm{ID}}^{\mathrm{DWD}}$ [hPa] |
|---|---|---|---|---|
| TK | 1.000104 | 0.999869 | 0.999973 | 0.027 |
| ETL | 0.999996 | 0.999869 | 0.999865 | 0.135 |
| WG | 1.000037 | 0.999869 | 0.999906 | 0.094 |

For Wollongong the WG-LR data are suffering from the low sample frequency and hence are not able to resolve the high
temporal variability of XCO. This can be seen nicely for the data recorded at 2022-12-23. There, a large peak in the XCO data is visible. However, the low-sampled WG-LR data are not able to sample this peak appropriately. Hence, when comparing data with very different sampling rates this can cause large differences.

    **Pressure Data** The pressure data collected at each site are summed up here and compared to the DWD Rheinstetten data by multiplying the bias compensation factors: $k_{\mathrm{ID_P}}^{\mathrm{DWD_P}} = k_{\mathrm{TS_P}}^{\mathrm{DWD_P}} \cdot k_{\mathrm{ID_P}}^{\mathrm{TS_P}}$. Assuming a pressure value of 1000 hPa the factors are
used to calculate an absolute difference in hPa by $\Delta_{\mathrm{ID}}^{\mathrm{DWD}} = 1000 \cdot \left(1 - k_{\mathrm{ID_P}}^{\mathrm{DWD_P}}\right)$. The largest deviation is found at the ETL site with a deviation of 0.135 hPa which is still a very low deivation. Hence, all sensors show an excellent agreement.

The pressure analysis is very important as it revealed the issues with the assumed height of the TCCON site in Wollongong as well as the not applied height correction for the TCCON analysis. An altitude of 5 m leads to a pressure difference of approximately 0.58 hPa. A study of Tu (2019) using PROFFIT as an evaluation software with low-resolution spectra, a change
of 1 hPa in the measured ground pressure causes an average increase of about 0.035% in $XCO_2$, 0.039% in $XCH_4$ and 0.052% in XCO, respectively. According to the measured level of pressure deviations, we do not expect them to have a large influence on the XGas values.

## 8    Conclusions

In this paper we successfully demonstrated the usage of an EM27/SUN as an international Travel Standard (TS) for the TC-
CON network. It was deployed to four TCCON sites on different continents: Tsukuba in Japan, East Trout Lake in Canada, Wollongong in Australia and Karlsruhe in Germany. Karlsruhe is the home base of the TS instrument and hosts the COCCON reference spectrometer. Therefore, the TCCON site Karlsruhe has been chosen as a reference for relative comparisons.

    Before and after each campaign at a TCCON site, the TS performed side-by-side measurements with the COCCON reference spectrometer located in Karlsruhe and the co-located TCCON-Karlsruhe instrument. Using these data bias compensation
factors are calculated to tie the TS instrument to the Karlsruhe TCCON site.

    At each site the TS measured side-by-side with the TCCON instrument for several days. In the period the TS was visiting a TCCON site, the TCCON instrument measured two different data products in an alternating way: The standard high-resolution TCCON data (XX-HR) and low-resolution data (XX-LR) with a maximal optical path difference of 1.8 cm matching the resolution of the EM27/SUN. For both data products a bias correction factor to the TS was calculated. By multiplying those





correction factors with the correction factors tying the TS to the COCCON reference, the visited TCCON sites are compared to the Karlsruhe TCCON site as a common reference.

At the Tsukuba site, a systematic error of the timestamp of the recorded interferograms could be found to be $-44$ s during the campaign. For the TK-LR data this error could be corrected and the analysis is carried out for both, the corrected and uncorrected TK-LR data. In Tsukuba as well as in Wollongong, high noise was found for $XGas$ products which was traced

back to low signal level in the spectral $O_2$ window. In East Trout Lake, Canada an important part of the TCCON instrument broke in the night before the TS arrived. Consequently, it was not possible to do a quantitative comparison.

The agreement found in Tsukuba and Wollongong for $XCO_2$ is on the 0.1% level whereas the time corrected low-resolution data show a higher deviation. For $XCH_4$ the agreement is within 0.2%, which also is a very satisfying result. For both the Tsukuba and the Wollongong data, the low-resolution $XCH_4$ data are biased low compared to the high-resolution data. This is

an interesting issue to be investigated in future campaigns.

For XCO, the deviations are larger than the TCCON requirements (several %) and are less consistent. However, the comparison of the Tsukuba data seem to suffer from unrealistic a priori profiles. The WG-LR data suffer from a low sampling frequency which probably causes the large differences and hence can not sample structure like the large peak at 2022-12-23 accurately. A summation of the results is given in Figure 16.

The TS is equipped with a pressure station which allows us to compare the pressure records of the different TCCON sites to a pressure stations at the German Weather Service (DWD) which is used for the Karlsruhe TCCON evaluation. The bias compared to the DWD station is $0.027$ hPa for the Tsukuba pressure records, $0.136$ hPa for the ETL pressure records and $0.094$ hPa for the Wollongong pressure records. In Wollongong the comparison of the pressure measurements revealed an error in the assumed heights of the sensor and the tracker which will be corrected in the official GGG2020 data.

For future campaigns several lessons can be learned from this study: pressure measurements shall not be measured inside a box with a venting fan (this issue was addressed after the Japan campaign). The TS requires a close monitoring of instrumental performance between deployments. The observation periods on site need to span sufficient time periods to reduce the random error budget. The XCO performance of the TS needs further evaluation.

*Code availability.* The PROFFAST sofware is available at https://www.imk-asf.kit.edu/english/3225.php. The PROFFASTpylot is available

at https://gitlab.eudat.eu/coccon-kit/proffastpylot The GGG2020 code is available at https://github.com/TCCON/GGG.

*Data availability.* Data are available on request from the corresponding author.



## Appendix A: Determination of Bias Compensation Factors

To compare the XGas results of two different spectrometers, in this work empirical relative bias factors are established. We assume these are air mass independent. They are used to describe the difference of a species XGas of instrument xx to the instrument yy regarded as reference and is denoted as $K_{xx}^{yy}(\text{XGas})$.

The procedure for all bias-compensation factors calculated in the course of this paper is always identical.

First, the data are filtered as described in Chapter 2.2.2. To derive the factors, the filtered XGas values of both instruments xx and yy are binned in intervals of $l$ minutes, denoted as $\overline{\text{XGas}}_{xx}^{t_i}$, where $t_i$ is enumerating the bins. Considering all coincident bins of both instruments the bias compensation factor is calculated by dividing the values of instrument yy by the ones of instrument xx and computing the average,

$$K_{xx}^{yy} = \frac{1}{N} \sum_{i=\text{coincident bins}}^{N} \frac{\overline{\text{XGas}}_{yy}^{t_i}}{\overline{\text{XGas}}_{xx}^{t_i}} \tag{A1}$$

$$= \frac{1}{N} \sum_{i=\text{coincident bins}}^{N} (q_{xx}^{yy})^i. \tag{A2}$$

Here, $(q_{xx}^{yy})_i = \frac{\overline{\text{XGas}}_{yy}^{t_i}}{\overline{\text{XGas}}_{xx}^{t_i}}$.

## A1  Error Analysis of the Bias Compensating Factors

The error of a measurement can be split into a systematic and a random error. Under constant conditions a systematic error falsifies repeated measurements by the same amount. In contrast, a random error is randomly influencing the results. The systematic errors of the TCCON stations and the TS give rise to the detected biases as described by the bias compensation factors. Here, we consider the random errors, which limits our ability to determine the correct bias compensation factors from a limited number of measurements. The random error is described by the standard error,

$$s_i^{\text{ID}} = \frac{\sigma_i^{\text{ID}}}{\sqrt{n_i^{\text{ID}}}} \tag{A3}$$

of the data, with $\sigma_i^{\text{ID}}$ the estimated standard deviation of the data of an instrument ID and $n_i$ the number of measurements in bin $i$. When binning the data in the intervals of $l$ minutes, we compute the standard error for each instrument in each l-minute bin.

The Gaussian error propagation of the relative error for the case of a quotient $x = \frac{x_1}{x_2}$ or a product $x = x_1 \cdot x_2$ is (see e.g. Kaloyerou (2018)),

$$\frac{\epsilon(x)}{x} = \left[ \left( \frac{\epsilon(x_1)}{x_1} \right)^2 + \left( \frac{\epsilon(x_2)}{x_2} \right)^2 \right]^{\frac{1}{2}}. \tag{A4}$$

Here, $\epsilon(x)$ describes the error on a quantity $x$. We use this notation to indicate that the error is not equal to the standard deviation and also to avoid it to be confused with the difference of two values which is denoted as $\Delta$.



Using Equation A4 the relative error for the quotient of the $i^{\text{th}}$ bin is calculated by,

$$\frac{\epsilon(q_{\text{xx}}^{\text{yy}})_i}{(q_{\text{xx}}^{\text{yy}})_i} = \left[ \left( \frac{s_i^{\text{xx}}}{\overline{\text{XGas}_{\text{xx}}}^{t_i}} \right)^2 + \left( \frac{s_i^{\text{yy}}}{\overline{\text{XGas}_{\text{yy}}}^{t_i}} \right)^2 \right]^{\frac{1}{2}} . \tag{A5}$$

The error of the final bias compensation factor is calculated using the Gaussian error propagation of Equation (A2).

$$\epsilon(K_{\text{xx}}^{\text{yy}}) = \left[ \frac{1}{N^2} \sum_{i=1}^{N} \left( \epsilon(q_{\text{xx}}^{\text{yy}})_i \right)^2 \right]^{\frac{1}{2}} . \tag{A6}$$

Note that because Equation (A2) is a sum the absolute errors $\epsilon(q_{\text{xx}}^{\text{yy}})_i$ and not the relative errors $\frac{\epsilon(q_{\text{xx}}^{\text{yy}})_i}{(q_{\text{xx}}^{\text{yy}})_i}$ are used. However, the errors given in the paper are the relative errors which are, $\frac{\epsilon(K_{\text{xx}}^{\text{yy}})}{K_{\text{xx}}^{\text{yy}}}$.

## A2 Tabulated Bias Compensation Factors

To compare a visited TCCON site with the reference in Karlsruhe, several the bias compensation factors between different instruments must be multiplied. This is depicted in Figure 15. In the main part of this paper, only the resulting bias of the TCCON sites relative to the reference in Karlsruhe ($K_{\text{XX}-\text{HR/LR}}^{\text{SN37}}$) in percentage are given. The intermediate bias compensation factors between the visited TCCON sites and the TS are given in Table A1.

The "virtual" bias compensation factors comparing the visited TCCON sites to the reference in Karlsruhe are given in Table A2. Based on these numbers the deviations in percentage which are given in Table 4 are calculated.

**Table A1.** The bias compensation factors for the TCCON-HR, and -LR data of the TK and WG sites to the TS (SN39). For the Tsukuba data also the time corrected LR data are given. The XX stands for the two letter TCCON-ID.

| Site | Species | $K_{\text{XX}-\text{LR}}^{\text{SN39}}$ | $K_{\text{XX}-\text{HR}}^{\text{SN39}}$ |
|------|---------|------------------|------------------|
| TK | $XCO_2$ | $1.00001 \pm 0.00007$ | $1.00087 \pm 0.00007$ |
| | $XCH_4$ | $1.00154 \pm 0.00008$ | $0.99771 \pm 0.00008$ |
| | XCO | $0.98673 \pm 0.00036$ | $0.93204 \pm 0.00031$ |
| TK | $XCO_2$ | $1.00000 \pm 0.00007$ | − |
| t-corr | $XCH_4$ | $1.00150 \pm 0.00008$ | − |
| $-44$s | XCO | $0.98688 \pm 0.00037$ | − |
| WG | $XCO_2$ | $1.00026 \pm 0.00007$ | $1.00037 \pm 0.00010$ |
| | $XCH_4$ | $1.00026 \pm 0.00008$ | $0.99872 \pm 0.00009$ |
| | XCO | $1.05846 \pm 0.00258$ | $0.98153 \pm 0.00105$ |

 

**Table A2.** The table shows the bias compensation factors between the visited TCCON sites and the Karlsruhe reference (SN37). All values are given with a random error followed by the calibration uncertainty. Both values are described in Section 7.2. Since the COCCON network as a whole is calibrated in a way that the reference spectrometer matches with the TCCON Karlsruhe data, a comparison with the COCCON reference spectrometer is equal to a comparison of the TCCON-Karlsruhe site.

| Site | Species | $K_{\text{XX-LR}}^{\text{SN37}}$ | $K_{\text{XX-HR}}^{\text{SN37}}$ |
|---|---|---|---|
| TK | $XCO_2$ | $0.99887 \pm 0.00008 + 0.00063$ | $0.99970 \pm 0.00008 + 0.00063$ |
| | $XCH_4$ | $1.00189 \pm 0.00009 - 0.00067$ | $0.99802 \pm 0.00009 - 0.00067$ |
| | XCO | $0.98832 \pm 0.00047 - 0.00053$ | $0.93383 \pm 0.00043 - 0.00050$ |
| TK | $XCO_2$ | $0.99836 \pm 0.00008 + 0.00063$ | – |
| t-corr | $XCH_4$ | $1.00112 \pm 0.00009 - 0.00067$ | – |
| $-44s$ | XCO | $0.98556 \pm 0.00047 - 0.00053$ | – |
| WG | $XCO_2$ | $0.99987 \pm 0.00007 + 0.00071$ | $0.99998 \pm 0.00010 + 0.00071$ |
| | $XCH_4$ | $1.00093 \pm 0.00008 - 0.00071$ | $0.99939 \pm 0.00010 - 0.00071$ |
| | XCO | $1.05909 \pm 0.00259 - 0.00622$ | $0.98212 \pm 0.00108 - 0.00577$ |

## Appendix B: Virtual Bias Compensation Factors and their Error Analysis

### B1 Virtual Bias Compensation Factors

The "virtual" bias compensation factors to compare the visited TCCON sites with the Karlsruhe reference are calculated by the multiplication of the factors between the TCCON site and the TS and the TS and the Karlsruhe reference:

$$K_{\text{TC-ID}}^{\text{TC-KA}} = K_{\text{TC-ID}}^{\text{TS}} \cdot K_{\text{TS}}^{\text{SN37}} \cdot K_{\text{SN37}}^{\text{TC-KA}} . \tag{B1}$$

To calculate a deviation in percentage, first an offset in units of a the column-averaged, dry air mole fraction is calculated. For this, the factors are multiplied by the average of the XGas over the whole period of a campaign $\overline{\text{XGas}}_{\text{TC-ID}}$,

$$\Delta \text{XGas}_{\text{TC-ID}}^{\text{TC-KA}} = \overline{\text{XGas}}_{\text{TC-ID}}(1 - K_{\text{TC-ID}}^{\text{TC-KA}}) . \tag{B2}$$

Using (B2), a deviation in percentage relative to the Karlsruhe TCCON site can be calculated using,

$$\Delta_\% \text{XGas}_{\text{TC-ID}}^{\text{TC-KA}} = \frac{\Delta \text{XGas}_{\text{TC-ID}}^{\text{TC-KA}}}{\overline{\text{XGas}}_{\text{TC-KA}}} \cdot 100 \tag{B3}$$

$$= \frac{1 - K_{\text{TC-ID}}^{\text{TC-KA}}}{K_{\text{TC-ID}}^{\text{TC-KA}}} \cdot 100 . \tag{B4}$$

With $\overline{\text{XGas}}_{\text{TC-KA}} = \overline{\text{XGas}}_{\text{TC-ID}} \cdot K_{\text{TC-ID}}^{\text{TC-KA}}$ the temporal mean of the KA data expressed using the correction factors and the temporal mean of the corresponding site. Equation (B4) is used to calculated the deviations given in Table 4.

### B2 Error Analysis

In this section the details of the error analysis of the virtual bias compensation factors are carried out.




**Random Error:** The first part describes the propagation of the individual random errors described in A1 when multiplying different bias compensation factors. In this case, the random error of the resulting product is calculated using Gaussian error propagation, as described in Equation (A4)

$$
\frac{\epsilon_{\mathrm{rand}}(K_{\mathrm{TC-ID}}^{\mathrm{SN37}})}{K_{\mathrm{TC-ID}}^{\mathrm{SN37}}} = \left[ \left( \frac{\epsilon(K_{\mathrm{TC-ID}}^{\mathrm{TS}})}{K_{\mathrm{TC-ID}}^{\mathrm{TS}}} \right)^2 \right.
$$
$$
\left. + \left( \frac{\epsilon(K_{\mathrm{TS}}^{\mathrm{SN37}})}{K_{\mathrm{TS}}^{\mathrm{SN37}}} \right)^2 \right]^{\frac{1}{2}} . \tag{B5}
$$

The errors described by Equation (B5) are given in Table A2.

When calculating the deviation in percentage, as given in Table 4, the relative random error is calculated by,

$$
\frac{\epsilon_{\mathrm{rand}}(\Delta_{\%}\mathrm{XGas})}{\Delta_{\%}\mathrm{XGas}} = \left[ \left[ \frac{\partial}{\partial K_{\mathrm{TC\text{-}ID}}^{\mathrm{SN37}}} (\Delta_{\%}\mathrm{XGas}_{\mathrm{TC\text{-}ID}}^{\mathrm{SN37}}) \right.\right.
$$
$$
\left.\left. \cdot \frac{\epsilon_{\mathrm{rand}}(K_{\mathrm{TC-ID}}^{\mathrm{SN37}})}{K_{\mathrm{TC-ID}}^{\mathrm{SN37}}} \right]^2 \right]^{\frac{1}{2}}
$$
$$
= \frac{1}{(K_{\mathrm{TC\text{-}ID}}^{\mathrm{SN37}})^2} \cdot \frac{\epsilon_{\mathrm{rand}}(K_{\mathrm{TC-ID}}^{\mathrm{SN37}})}{K_{\mathrm{TC-ID}}^{\mathrm{SN37}}} . \tag{B6}
$$

**Calibration uncertainty:** The second part is the uncertainty introduced by a potential drift of the TS instrument relative to the COCCON reference. Its upper limit is estimated by using the $\Delta K_{\mathrm{SN39}}^{\mathrm{SN37}}$ of the bias compensation factors measured before and after each campaign as given in Table 1. Since $\Delta K_{\mathrm{SN37}}^{\mathrm{SN37}}$ are values in percentage, the uncertainty of the final $K_{\mathrm{ID\text{-}LR/HR}}^{SN37}$ are calculated by,

$$
\epsilon_{\mathrm{cal}}(K_{\mathrm{XX\text{-}LR/HR}}^{\mathrm{SN37}}) = \Delta K_{\mathrm{SN39}}^{\mathrm{SN37}} \cdot K_{\mathrm{XX\text{-}LR/HR}}^{\mathrm{SN37}} \cdot \frac{1}{100} \tag{B7}
$$

In Table A2 the uncertainty is given as the second value.

For the deviation in percentage, the calibration uncertainty is calculated using linear error propagation of Equation (B4) and using the error given in Equation (B7),

$$
\epsilon_{\mathrm{calib.}}(\Delta_{\%}) = \frac{\partial}{\partial K_{\mathrm{xx}}^{\mathrm{yy}}} \Delta_{\%}\mathrm{XGas}_{\mathrm{xx}}^{\mathrm{yy}} \cdot \Delta K_{\mathrm{xx}}^{\mathrm{yy}} \tag{B8}
$$
$$
= \frac{-1}{(K_{\mathrm{xx}}^{\mathrm{yy}})^2} \cdot \epsilon_{\mathrm{calib}}(K_{\mathrm{XX\text{-}LR/HR}}^{\mathrm{SN37}}) \cdot 100 . \tag{B9}
$$

The result of this error analysis is given in Table 4. Here again, the random error is given first with a ± sign and the calibration uncertainty is given second.

## Appendix C: Raw Data of Tsukuba and Wollongong with Time Correction

The airmass dependency of XAIR found in the Tsukuba TCCON data can be traced back to a wrong timestamp of the spectra. For the LR data, empirically it is found that an offset of $-44\,\mathrm{s}$ can correct the airmass dependency. The corrected data are shown



in Figure C1. Note, that this is no official TCCON data. For the TK-HR data, the timing error is currently under investigation. As soon as it is solved, the data will be submitted to TCCON.

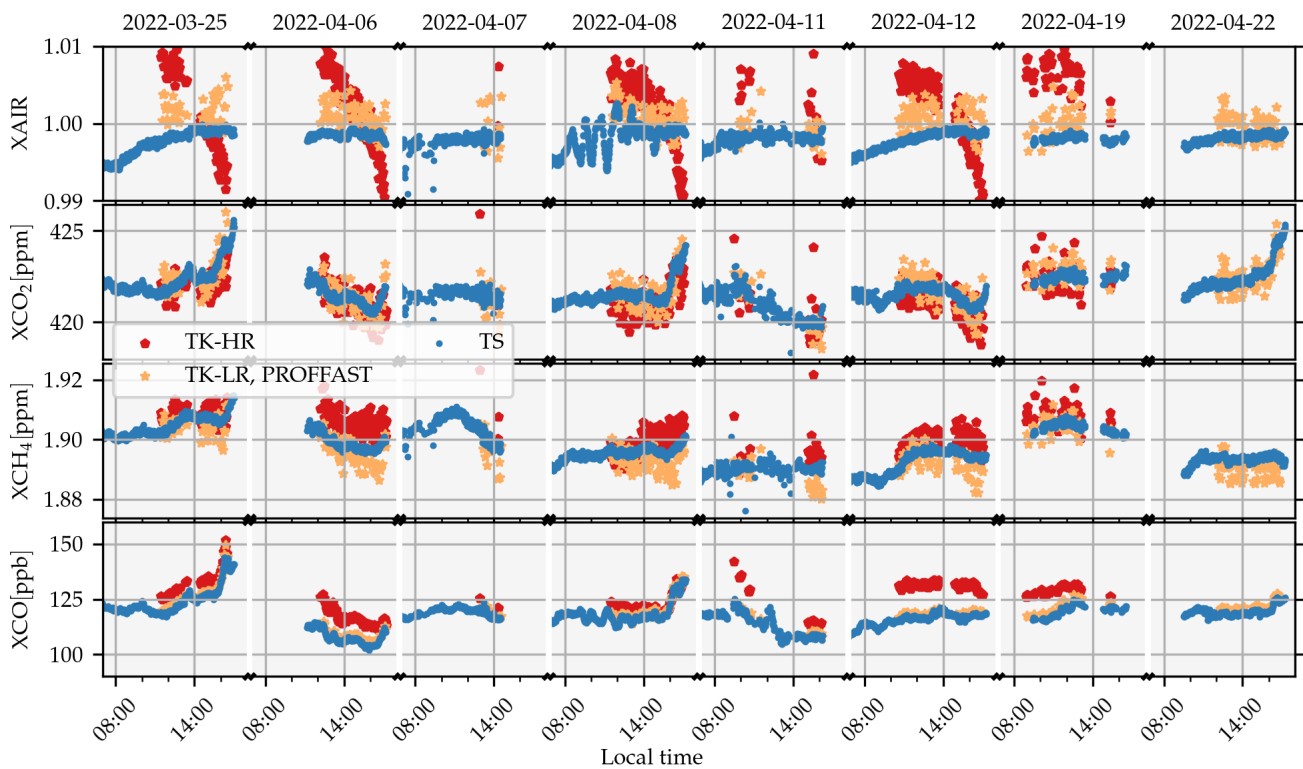

**Figure C1.** The XGas results for $XCO_2$, $XCH_4$ and $XCO$ of the side-by-side measurements in Tsukuba, Japan with a time correction of $-44$ s for the LR data. This removes the time dependency. Note, that the data shown here is no official TCCON data, as the time error is going to be corrected before publishing it.

### Appendix D: Pressure Plots

In Figure D1, D2 D3 the comparison of the pressure measurements of the TCCON sites in Tsukuba, ETL and Wollongong with the TS using a Vaisala PTB330 and the TCCON pressure sensors are plotted. The data is discussed in the main text.

*Author contributions.* **Benedikt Herkommer** coordinated the deployment of the TS to the different stations, collected measurements, did the data analysis, wrote the manuscript, did modifications on the Hardware of the enclosure, contributed to the PROFFAST development. **Carlos Alberti** performed ILS measurements in Karlsruhe. **Paolo Castracane** and **Angelika Dehn** regularly monitored the progress of the

TS and provided telecons for the discussions with other scientists. **Matthias Max Frey** set up the instrument in Tsukuba and collected



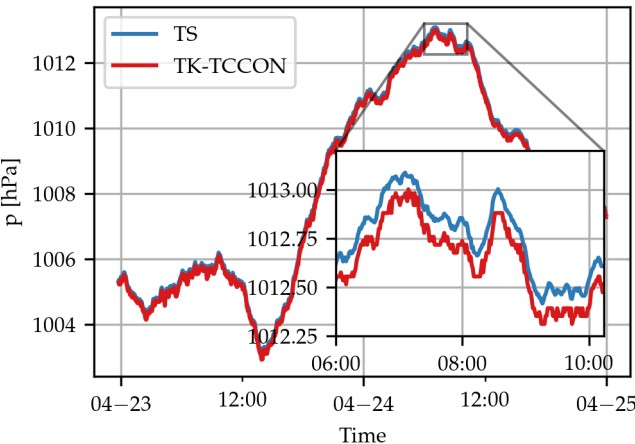

**Figure D1.** The pressure recorded at the Tsukuba TCCON site with the official TCCON sensor (TK) is plotted in orange and the TS in blue. From the inset one can see that there is a small difference of $-0.105$ hPa on average. This results in a bias compensation factor of $k_{\mathrm{TK_P}}^{\mathrm{TS_P}} = 1.000104$. For the comparison the TS pressure sensor was placed side-by-side at the same height as the TK pressure sensor.

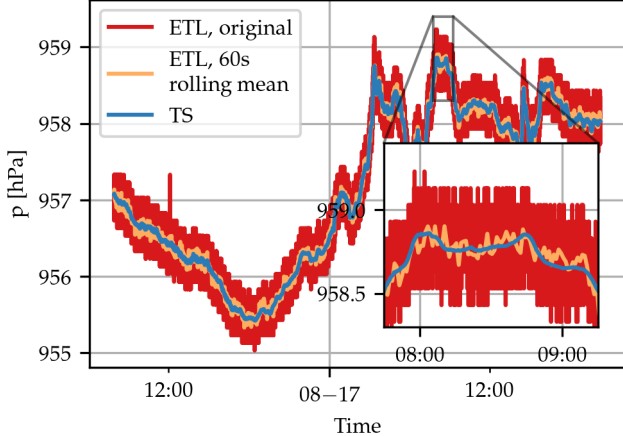

**Figure D2.** The pressure recorded at the ETL-TCCON site with the official TCCON sensor and the TS. The TCCON data show a high noise level. This is accounted for by taking the rolling mean with a window size of $60$ s, plotted in orange. The original data are plotted in green. For the comparison both, the TS and the rolling mean data are resampled to $60$ s bins. This yields a average deviation of $-0.00419$ hPa and a bias compensation factor of $k_{\mathrm{ETL_P}}^{\mathrm{TS_P}} = 0.999996$.



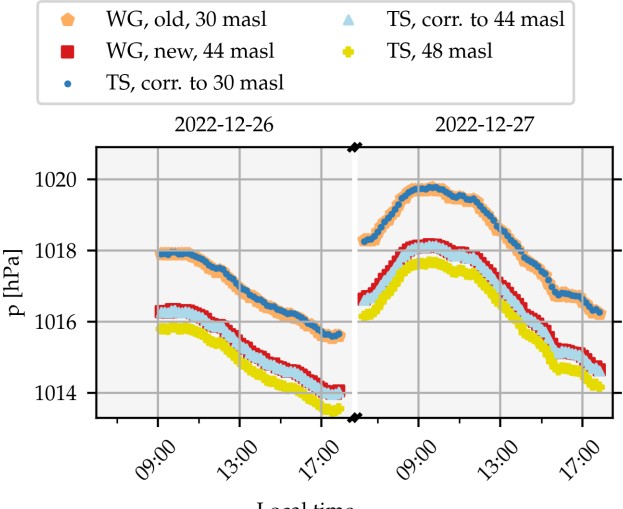

**Figure D3.** Pressure comparison for two exemplary days of the Wollongong pressure sensors and the TS. In WG, there is an old and a new TCCON spectrometer. The old one is the operable one, whereas the new one is still in the testing phase. Both are equipped with pressure sensors. The old sensor is at 30 masl, the new is at 44 masl. The TS measured at 48 masl altitude and hence its data are height corrected by $-4$ m (purple dots) and $-18$ m (blue dots). For both TCCON pressure sensors the data are in good agreement with the height corrected TS data. On average the pressure sensors of the new and old TCCON site deviates by $0.02517$ hPa and $-0.03770$ hPa relative to the TS pressure measurements height corrected by $-4$ masl and $-18$ masl. For the pressure sensor of the old TCCON site this gives a pressure compensation factor of $K_{\mathrm{WG_p}}^{\mathrm{TS}} = 1.0000373$. The shown days are chosen randomly, the numbers are calculated using the whole pressure record available.

measurements. **Isamu Morino** collected TCCON measurements at Tsukuba. **Nasrin Mostafavi Pak** collected measurements at East Trout Lake. **Lawson Gillespie** collected measurements at East Trout Lake. **Debra Wunch** contributed to the GGG2020 development and collected measurements at the East Trout Lake TCCON site. **Florain Dietrich** built the enclosure. **Jia Chen** created the study design for the enclosure. **Nicholas Deutscher** collected measurements with the TS and TCCON at Wollongong. **Brittany Walker** performed the GGG2020 evaluation

of the Wollongong data. **Jochen Groß** developed software for remote access, did hardware work on the enclosure. **Frank Hase** created the study design, developed PROFFAST. All co-authors provided feedback on the manuscript.

*Competing interests.* Frank Hase is a member of the editorial board of Atmospheric Measurement Techniques.

*Acknowledgements.* We thank ESA for the funding of activities in support of COCCON by KIT in the context of the projects FRM4GHG-II, COCCON-PROCEEDS, COCCON-OPERA and QA4EO. The operation at the Tsukuba TCCON site is supported in part by the GOSAT

series project. The TCCON station at ETL is supported by the Canada Foundation for Innovation, the Ontario Research Fund, and Environ-



ment and Climate Change Canada. JC and FD acknowledge the funding from the German Research Foundation DFG (CH 1792/2-1; INST 95/1544).



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
