# Peer review of "Using a portable FTIR spectrometer to evaluate the consistency of TCCON measurements on a global scale: The COCCON Travel Standard"

_EGUsphere, 2023_

## Author Comment (AC1)

**Answers to the comments of "Anonymous Referee 2"**

First of all, we would like to thank you for the valuable feedback and comments.

In the following we will reply to your questions and comments point-by-point. Your questions and comments are given (sometimes a bit shortened) in bold font, and our answers in normal font. Extracts of the paper are added in italic font.

Note that two errors were discovered by us in the meanwhile which also have been corrected:

1. In Table 1 and Table A2 there were wrong numbers. Fortunately, only the numbers in the manuscript were erroneous but not the ones used for the calculation of the final result. Hence, nothing changed in Table 4 due to these corrections.
2. We discovered an error in the Excel table used to calculate the final corrections factors for the Tsukuba time-corrected data. This error is corrected now, causing that the relative deviation for the $XCO_2$ TK-LR-tcorr data is now similar to the $XCO_2$ TK-LR data.
   This caused the values for "TK t-corr -44s" to be changed in Table 4 and Table A2:
   a. Table 4:
      $XCO_2$: 0.16401 ± 0.00830 − 0.06318 → 0.11387 ± 0.00829 − 0.06314
      $XCH_4$: −0.1115 ± 0.00873 + 0.06690 → −0.18343 ± 0.00871 + 0.06685
      $XCO$:  1.46537 ± 0.0487 + 0.05470 → 1.16653 ± 0.04870 + 0.05454
   b. Table A2:
      $XCO_2$: 0.99836 ± 0.00008 + 0.00063 → 0.99886 ± 0.00008 + 0.00063
      $XCH_4$: 1.00112 ± 0.00009 − 0.00067 → 1.00185 ± 0.00009 − 0.00067
      $XCO$: 0.98556 ± 0.00047 − 0.00053 → 0.98847 ± 0.00047 − 0.00053
   This also caused changes in Figure 16 (which is the visualization of Table 4) and the main text in Section 7.3:

Furthermore, we corrected grammatic and spelling errors as well as the wrong naming of the colors and shapes in the main text.

**Specific Comments**

1. **Is there a time series of the transport logger?**

   o No, only shocks are recorded, however, including a timestamp. To make this clearer, we described this in the manuscript and added the start and stop dates of the logging as well as the timestamp of the detected shock events.
   o We added in the manuscript in Section 2.2.1:
   *"The loggers do not record a continuous time series but only log shocks with a duration and acceleration larger than a certain threshold. Furthermore, the sensors are saturated a 16 g. Hence, all shock events larger than that are truncated to 16 g."*
   *"On its way to Wollongong, the record was started on 2022-10-22 at 07:59 (this and all following times are given in UTC) and stopped on 2022-12-06 at 09:35. The events were recorded on 2022-11-25 08:23 and 10:34 as well as on 2022-12-06 at 09:26. On its back, the record started on 2023-01-26 at 21:27 and stopped on 2023-11-07 at 11:32. The event was recorded on 2023-02-15 at 03:40."*
   *"On its way to Wollongong the logger attached to the EM27/SUN was started on 2022-10-20 at 7:59 and stopped on 2022-12-06 at 09:40.*

*It recorded one shock event on 2022-12-06 at 09:40 with a maximum acceleration of 14.4 g. Since this record was just before stopping the record, this was probably caused by putting the logger hardly on the desk before reading it out."*
*"On its way back the record starts on 2023-01-26 at 21:35 and stopped on 2023-03-07 at 11:33. Two shock events were recorded both on 2023-01-26 at 21:38 and a maximum acceleration of 16 g. Here, as well the record was shortly after the start and therefore is most probable caused by a drop of the logger itself without being attached to the instrument."*
*"The fact that the enclosure experienced such extreme shocks, but the logger attached to the EM27/SUN did not record them indicates that the packing in foam of the EM27/SUN helps to cushion the shocks".*

2. **Comparison with Saturn V is misleading as rocket launch acceleration is a steady state acceleration.**

   o Agreed, we removed the comment. (Very high variable accelerations seem to have resulted from extreme vibrations especially during the first stage burning of the Saturn V, according to Apollo mission astronauts' reports)

3. **Lines 20-23: Not clear if the comparisons described refer to TCCON HR-LR or TCCON-TS.**

   o These are the final deviations calculated (visited TCOCN site relative to the reference in Karlsruhe), so TCCON versus TS.
   o We reformulated these lines to make that clear:
   *"For Tsukuba and Wollongong the agreement with the reference in Karlsruhe found for $XCO_2$ is…"*

4. **Why not compare directly to the KA-TCCON site but to the COCCON reference EM27/SUN?**

   o As correctly stated by you, Sha et al. (2020) found a seasonal bias between HR and LR FTIR remote sensing measurements. This is triggered by the different vertical sensitivity of high-resolution and low-resolution measurements. Therefore, whenever the a-priori gas profile deviates from the actual profile, a difference between the XGas result occurs. The most prominent effect are the aforementioned seasonally varying biases.

   o However, for the TS characterization measurements conducted before and after each campaign we need to avoid these variable biases, because they would result in a time-dependent bias in the TS's calibration, which would propagate to the comparisons to the TCCON stations. Hence, we compare with low-resolution FTIR measurements.

   o A more direct approach would be to use LR measurements of the TCCON-KA site. However, the TCCON-KA spectrometer collects alternating HR and LR measurements and follows a measurement pattern which also involves the collection of mid-infrared spectra. As consequence, LR TCCON data are available only every 20 minutes. In contrast, we approximately collect one measurement every minute with the reference COCCON spectrometer. Hence, we can achieve significantly better statistics from the comparison of the TS with the COCCON reference spectrometer. The COCCON reference, operated in Karlsruhe continuously, can be compared to the TCCON-KA station record with much lower statistical uncertainty as it can be compared over a longer time interval.

   o The calibration factors within the PROFFAST retrieval software are determined such, that on average the COCCON reference agrees with the TCCON-KA-HR

measurements (see Alberti et al. (2022), Figure 20 and Herkommer (2023), Chapter 3). Therefore, it is justified (and the best strategy) to compare with the COCCON-reference spectrometer.

- o  To make this clearer in the manuscript we added the following to Section 3 of the manuscript:
  *"The reason why we are comparing to the COCCON reference and not directly to the TCCON-KA site is the following: As mentioned earlier, for short-term comparison different resolutions can induce variable biases in the final XGas products. To avoid these, it would still be possible to compare LR data measured with the TCCON-KA spectrometer with the TS. However, the focus of the TCCON-KA measurement is to collect standard TCCON and mid-infrared measurements with high resolution, hence, we only collect a LR spectrum every 20 minute. Therefore, there are significantly less TCCON-KA LR measurements available than measurement with the COCCON reference unit which collects about one measurement per minute. The airmass independent calibration factors used internally in the PROFFAST2 software are carefully chosen such, that the COCCON reference is tied to the official TCCON-KA HR data."*

5. **Is a seasonal bias as mentioned by Sha et al. (2020) considered?**

- o  This effect described by Sha et al. (2020) is an effect of the different spectral resolutions of the instruments (low-resolution portable versus high-resolution TCCON observations).
- o  To avoid this disturbance, our study design incorporates the TCCON-LR data for the site evaluation. This does not imply a loss of information, as the low-resolution TS does not provide any handle for the verification of the high-resolution part of the TCCON measurement. This aspect is covered by TCCON by using gas cell measurements instead.
- o  We also report the official TCCON-HR vs TS differences, because these are undoubtedly of interest. But this comparison is inherently "noisy" due to the variable smoothing error contributions resulting from the different vertical sensitivities of low and high-resolution measurements. We agree, that this fact was not carried by us properly in the manuscript so far. Therefore, we added the following:
- o  In Section 2.2.2:
  *"These effects are also observed by Sha et al. (2020)."*
  and
  *"As a consequence of the different resolutions it is important to note that the comparison of the TCCON-HR data with the TS data are affected by variable smoothing error contributions resulting from the different vertical sensitivities of low and high-resolution measurements. The judgement of the level of agreement of the TS measurements with the TCCON site measurements needs to be based on the TCCON-LR data. This does not imply a loss of information, as the low-resolution TS measurement does not provide any handle for verifying the high-resolution part of the TCCON measurement. This latter aspect needs to be checked by the use of low-pressure gas cells. Once the TS has visited a larger number of sites, a larger dataset of TCCON-HR vs TS comparison is available. This can probably be used to see systematic effects of over-, or underestimation of different gases by the different resolutions."*
- o  In Section 7.3:
  *"For the following discussion it is important to keep in mind that the comparison of the HR data are affected by variable smoothing error contributions resulting from the different vertical sensitivities of low and high-resolution measurements. This introduces an uncertainty when comparing XGas results."*

**Technical corrections**

1. **L43: has been evaluated**

   o Done.

2. **L51: omit "profile observations by"**

   o "collocated airborne profile observations" --> "collocated airborne measurements"

3. **L208 even -> event**

   o Changed this paragraph, therefore removed.

4. **L273: remove first limits**

   o Done.

5. **L295: and Fig 2 caption Red crosses should be blue triangles.**

   o Done.

6. **Figure 4. Delta XCH4 is presented, relative to what? (Ref instrument is assumed)**

   o Yes, to the COCCON reference instrument. This is added to the manuscript:

   *"Investigating the dependency of $\Delta XCH_4$ $\Delta XCH_4^{S5P}$ of the reference EM27/SUN and the TS device as a function of the solar zenith angle (SZA)."*

7. **Figure 6. Although not necessary in this case it is generally more pleasing to have the same y axis scale on adjacent plots.**

   o Done.

8. **L407: Extra the**

   o Done.

9. **Figure 10 caption. Repeated use of "normed" throughout. It is more normal to use normalized.**

   o Corrected.

10. **Fig 16 caption. Last sentence redundant, every figure should be discussed in the man text.**

    o Removed.

11. **L662: l-minute or one-minute?**

    o l-minute. Set "l" in math mode to make it clearer.

12. **Equations B7-B9 check the subscripts for consistency**

    o Done.

References:

Alberti, C. et al: "Intercomparison of low- and high-resolution infrared spectrometers for ground-based solar remote sensing measurements of total column concentrations of CO2, CH4, and CO", Atmospheric Measurement Techniques, 13, 2022, DOI: 10.5194/amt-13-4791-2020

Herkommer, B. 2023: "Improving the consistency of greenhouse gas measurements from ground-based remote sensing instruments using a portable FTIR spectrometer", Dissertation, Karlsruher Institut für Technologie (KIT), DOI: 10.5445/IR/1000168723

---

## Author Comment (AC2)

**Answers to the comments of "Anonymous Referee 1"**

First of all, we would like to thank you for the valuable feedback and comments.

In the following we will reply to your questions and comments point-by-point. Your questions and comments are given (sometimes a bit shortened) in bold font, and our answers in normal font. Extracts of the paper are added in italic font.

Note that two errors were discovered by us in the meanwhile which also have been corrected:

1. In Table 1 and Table A2 there were wrong numbers. Fortunately, only the numbers in the manuscript were erroneous but not the ones used for the calculation of the final result. Hence, nothing changed in Table 4 due to these corrections.
2. We discovered an error in the Excel table used to calculate the final corrections factors for the Tsukuba time-corrected data. This error is corrected now, causing that the relative deviation for the $XCO_2$ TK-LR-tcorr data is now similar to the $XCO_2$ TK-LR data.
   This caused the values for "TK t-corr -44s" to be changed in Table 4 and Table A2:
   a. Table 4:
      $XCO_2$: 0.16401 ± 0.00830 − 0.06318 → 0.11387 ± 0.00829 − 0.06314
      $XCH_4$: −0.1115 ± 0.00873 + 0.06690 → −0.18343 ± 0.00871 + 0.06685
      $XCO$:  1.46537 ± 0.0487 + 0.05470 → 1.16653 ± 0.04870 + 0.05454
   b. Table A2:
      $XCO_2$: 0.99836 ± 0.00008 + 0.00063 → 0.99886 ± 0.00008 + 0.00063
      $XCH_4$: 1.00112 ± 0.00009 − 0.00067 → 1.00185 ± 0.00009 − 0.00067
      $XCO$: 0.98556 ± 0.00047 − 0.00053 → 0.98847 ± 0.00047 − 0.00053
   This also caused changes in Figure 16 (which is the visualization of Table 4) and the main text in Section 7.3:

**General Comments**

**How often should Travel Standard visits be conducted?**

- This is a good question and the ideal answer would be "as often as possible".

- More differentiated, it would be useful to re-visit sites after any major instrumental intervention, or as in case of the timing error in Tsukuba, after a recognized problem is solved (or reprocess the data if it is solved software-sided). Drifts or sudden changes in the XAIR time series of a TCCON site turn out to be good indicators for the occurrence of instrumental changes that would make another site visit desirable.

- However, as one can see, the TS activities are quite an effort: The study described in this article ran for 15 month and we visited 3 sites. This long-time span of in average 5 month per site is caused by a) the time needed for the shipment and customs procedures b) the time needed at each site to collect sufficient data and c) the time for maintaining and assuring the proper characterization of the TS at KIT. Assuming this as an average speed, it would take around 12 years to visit all TCCON sites, which is too long. To speed this up, there are several approaches.

  - The rigorous solution would be to use several EM27/SUN being as TSs which are all based in Karlsruhe. Already, the use of a pair of spectrometers sharing the same shelter already would speed up the procedure. Hence, it is possible to have one EM27/SUN at a

campaign with the shelter, and the other is simultaneously collecting side-by-side measurements in Karlsruhe without a shelter.

- o Furthermore, it would be possible to combine several sites (e.g. visit more than one sites in the US before returning to KA). However, operating the TS en route without re-characterization in between would increase the risk of drifts and reduce the success chances of prolonged campaigns.
- o Also, possible would be to use other EM27/SUNs to compare TCCON sites close by to each other (e.g. all sites in Japan) and only visit one of the sites with the TS.
- o Furthermore, the approach of cascading down the global inter-continental calibration achieved by the TS presented here looks very promising. The TS would then visit a certain TCCON site in a region and the occasion of this encounter would be used to collocate further COCCON spectrometers operated by groups of this region during the TS visit. These spectrometers are used subsequently to spread out the results to further TCCON sites nearby.

- These thoughts are now included in the outlook of the paper:

*"To make use of the valuable insights provided by the TS it would be desirable to visit TCCON sites regularly. However, the TS activities are quite some effort as it can be seen in this study. Continuing with the same speed would take around 10 years to visit all the TCCON sites (~ 3 per year). To speed this up different approaches are possible: The most direct one, which is already planned, would be to use several closely monitored EM27/SUNs to be used as TS in parallel sharing the same enclosure. This helps to increase the frequency of campaigns as one of the EM27/SUN spectrometers can be sent to a campaign whereas in parallel the other can perform side-by-side measurements in KA. Also, it would be possible to visit several sites between two calibration stops at KA. However, this would reduce the accuracy as the TS is less closely monitored. Another approach would be to visit one TCCON site per country and transfer its level to surrounding sites by using other EM27/SUN, which of course, must be monitored closely, too."*

**Ultimately, we need to tie all the measurements back to the WMO scale. If the KIT TCCON instrument is going to be the reference, should aircraft overflights or AirCore launches be conducted more regularly at the site to keep that instrument tightly related to the WMO scale? If, for logistical reasons, the KIT TCCON site isn't ideal for overflights/launches, could another site be used for this purpose and the Travel Standard relate that TCCON instrument to the KIT instrument?**

- We agree that it is of great importance to tie the TCCON to the WMO scale. However, the idea of the TS is not to improve the absolute calibration of the TCCON but to investigate and increase the consistency within the TCCON. Specifically, we try to reduce station-to-station biases across TCCON caused by instrumental imperfections of real-world spectrometers.

- To make this intention clearer we added the following to the introduction:

We changed "In order to produce reliable reference data, it is important to ensure that the network as a whole is accurately tied to the World Meteorological Organization's (WMO) trace gas scale (Hall et al., 2021, Dlugokencky et al., 2005), and that the network has minimal station-to-station biases."
to
"To produce reliable reference data, two things have to be considered. The first item is to ensure that the network as a whole is accurately tied to the World Meteorological Organization's (WMO) trace gas scale (Hall et al. (2021), Dlugokencky et al. (2005). The second is to minimize station-to-station biases across the network due to non-nominal behavior of the spectrometer."

Furthermore, we added (the addition is underlined, the italic text is given for context):

*"In this work an additional method of further enhancing the TCCON's quality management is presented and applied. It is based on a portable EM27/SUN FTIR spectrometer operated in the framework of the Collaborative Carbon Column Observing Network (COCCON) (Frey et al. (2019)) which will be used as a traveling standard. This activity aims directly at the improvement of the site-to-site consistency.*"

**Specific Comments**

1. **Define XGas earlier in the paper.**

   o Is now defined in the introduction.

2. **Line 23: Explain what kind of pressure is meant by "pressure analysis"**

   o Added "An important auxiliary value for FTIR retrievals is the surface pressure. Using the pressure sensor onboard the TS, the surface pressure measurements at each site are also compared. The surface pressure analysis reveals..." at line 23.

3. **Line 49/50: Provide reference for WMO trace gas scale.**

   o Added "Hall et al. 2021", https://doi.org/10.5194/amt-14-3015-2021 for CO2
   o Added "Dlugokencky et al. 2005": https://doi.org/10.1029/2005JD006035

4. **Lines 128 - 133: No additional info here than in the Intro. Trim the Intro.**

   o Deleted line 56 and following in intro: "However, the collection of such a profile data set is laborious, expensive and the number of available in-situ profiles is too small for detecting minor biases of individual TCCON sites. Moreover, TCCON sites located in populated regions with severe flight restrictions are particularly difficult to address with this strategy."

5. **Adding a subscript or similar to mark, that XAIR from PROFFAST and GGG are different.**

   o Added "To make this clear we add the subscript "GGG" to the XAIR labels to indicate that we are using the standard GGG XAIR values and the inverted PROFFAST XAIR values." to the main text.
   o Added XAIR_GGG to the labels of the figures and an explanation to the caption.

6. **Line 172-178: Redundant text earlier to this section Remove for brevity.**

   o These lines were intended to be a summary. They have changed to: "In summary, we believe that the COCCON-TS for the TCCON presented in this paper is a valuable complement to the methods presented above"

7. **Line 153 - 155: Provide units for all quantities.**

   o When possible units and quantities were added.

8. **Line 212 - 216: Description of the logging events is not clear.**

   o Reformulated these lines, added times of the logging events

9. **Line 218-219: Sentence is not necessary. Remove for brevity.**

   o Has been removed.

10. **Line 309: "It is assumed that SN37 is constant. How is this assumption justified?**

    o This is justified by a long-term analysis of the TCCON-KA and SN37 data as shown in Alberti et al. 2022: https://doi.org/10.5194/amt-15-2433-2022, Figure 20.
    o Added "This assumption is justified by a long-term analysis of the reference EM27/SUN spectrometer (SN37) with the TCCON-Karlsruhe data as shown in Alberti et al. 2022, Figure 20."

11. **Lines 342-349: This correction approach effectively assumes the dependence on SZA is 100% in the Travel Standard instrument. How does the uncertainty in this assumption propagate into the calculation of absolute uncertainty of the CO measurements?**

   o For all gases we use the side-by-side measurements before and after each campaign to derive an upper threshold for drifts of the XGas results during the campaign.
   o Hence, the smaller the differences of the TS to the reference before and after the campaign, the smaller the uncertainty.
   o With the CO correction, we try to match the COCCON SN37 reference data as closely as possible. Any deviation from this (i.e. an inaccuracy of the CO-correction) is captured by the difference of the bias-compensation K_SN39^SN37 CO before and after the campaign. This difference is then propagated into the uncertainty of the final result.
   o Hence, the used method of deriving the calibration uncertainties implicitly includes the uncertainty of the CO correction.

12. **Lines 353 - 354: Why is TCCON KA not using own p Sensor?**

   o This is for historical reasons but offers several advantages:
      ▪ Several comparisons with pressure sensors operated at KIT have proven that the DWD-sensors data agree with test-measurements at KIT (see also the comparison in this paper.)
      ▪ The DWD-sensor is part of the operational weather service network and therefore monitored closely and the data are quality checked. Hence, we assume that the surface pressure data are of reliable quality.
      ▪ Furthermore, we do not have to worry about maintaining and calibration of the sensor.

13. **Fig 8 Add to caption that TK-LR GGG only plotted for XCO.**

   o Added in the text: "Note that in Figure8 the GGG values are only plotted for XCO."
   o Added in the caption: "Note that for XCO the TK-LR data is also processed with GGG2020 and plotted using black triangles."

14. **General: Change caption and explanation of colors + shapes used in the plots**

   o Changed caption and text for Fig 1.
   o Caption Figure 5: Changed to "blue triangle shaped markers".
   o Caption Figure 8: Changed colors and added markers.
   o Caption Figure 9: Changed marker and colors.
   o Caption Figure 10: Changed colors
   o Adapted the text referring to Figure 10.
   o Section 6: Changed description of Figure 12.
   o Line 434: Yellow x-shaped markers --> black triangles
   o Lines 448, 449, 464: Colors in Figures do not match --> Now, they do
   o Line 512: WG-HR are red not green --> adapted to new colors.